# Pathogenic NLRP3 mutants form constitutively active inflammasomes resulting in immune-metabolic limitation of IL-1β production

Cristina Molina-López[1], Laura Hurtado-Navarro[1], Carlos J. García[2], Diego Angosto-Bazarra[1], Fernando Vallejo [2], Ana Tapia-Abellán[1,9], Joana R. Marques-Soares[3], Carmen Vargas[4], Segundo Bujan-Rivas[3], Francisco A. Tomás-Barberán [2], Juan I. Arostegui [5,6,7] & Pablo Pelegrin [1,8] ✉

Cryopyrin-associated periodic syndrome (CAPS) is an autoinflammatory condition resulting from monoallelic *NLRP3* variants that facilitate IL-1β production. Although these are gain-of-function variants characterized by hypersensitivity to cell priming, patients with CAPS and animal models of the disease may present inflammatory flares without identifiable external triggers. Here we find that CAPS-associated NLRP3 variants are forming constitutively active inflammasome, which induce increased basal cleavage of gasdermin D, IL-18 release and pyroptosis, with a concurrent basal pro-inflammatory gene expression signature, including the induction of nuclear receptors 4 A. The constitutively active NLRP3-inflammasome of CAPS is responsive to the selective NLRP3 inhibitor MCC950 and its activation is regulated by deubiquitination. Despite their preactivated state, the CAPS inflammasomes are responsive to activation of the NF-κB pathway. NLRP3-inflammasomes with CAPS-associated variants affect the immunometabolism of the myeloid compartment, leading to disruptions in lipids and amino acid pathways and impaired glycolysis, limiting IL-1β production. In summary, NLRP3 variants causing CAPS form a constitutively active inflammasome inducing pyroptosis and IL-18 release without cell priming, which enables the host's innate defence against pathogens while also limiting IL-1β–dependent inflammatory episodes through immunometabolism modulation.

The Nucleotide-binding oligomerization domain, leucine-rich repeat receptor, and pyrin-domain containing-protein 3 (NLRP3) inflammasome is a multiprotein cytosolic complex involved in different human inflammatory diseases[1]. The NLRP3-inflammasome triggers the activation of caspase-1, which in turn processes several proteins within the cell, including gasdermin D (GSDMD) and the pro-inflammatory cytokines interleukin (IL)−1β and IL-18. The amino-terminal fragment of GSDMD inserts itself into the plasma membrane, oligomerizes and forms pores, which facilitate the release of IL-1β and IL-18, and initiate a specific type of cell death known as pyroptosis[2]. Rare variants in the *NLRP3* gene are responsible for a monogenic autoinflammatory disease called Cryopyrin-associated periodic syndrome (CAPS)[3,4]. CAPS includes three different clinical phenotypes with increasing severity: familial cold autoinflammatory syndrome (FCAS) as the mildest form,

the Muckle–Wells syndrome (MWS) as the intermediate phenotype, and the neonatal-onset multisystem inflammatory disease/chronic infantile neurologic cutaneous articular syndrome (NOMID/CINCA) as the most severe expression[4,5]. Typically, these phenotypes manifest early in life and are characterized by recurrent episodes of fever, urticaria-like skin rashes, conjunctivitis, and joint inflammation. From a genetic point of view, most CAPS patients carry germline *NLRP3* variants, with a smaller group carrying post-zygotic variants[6–8]. While specific *NLRP3* variants have been associated with each CAPS clinical phenotype, current understanding suggests that some *NLRP3* variants may cause overlapping phenotypes. This supports the idea that CAPS behaves more like a syndrome with a variable spectrum of features rather than a singular entity[5].

While *NLRP3* variants associated with CAPS are typically considered gain-of-function due to their dominant phenotype, most in vitro studies rely on the bacterial trigger lipopolysaccharide (LPS) to induce the activation of these inflammasomes. This activation is then detected by the release of IL-1β[9–16]. The canonical activation of the wild type NLRP3 inflammasome requires two sequential steps. The first step, or priming, induces nuclear factor (NF)-κB activation and transcriptional expression of *NLRP3* and *IL1B*. It also promotes key NLRP3 post-transcriptional modifications, notably deubiquitination[17,18]. The second step relies on cellular stress inducers, such as the decrease of intracellular $K^+$ by the ionophore nigericin or the activation of the cationic P2X7 receptor. Both of these triggers cause a conformational change in the NLRP3 inactive structure[19,20]. Therefore, NLRP3 with gain-of-function variants may only require priming to assemble an active inflammasome, rendering it hypersensitive. However, is not well understood how the CAPS-associated NLRP3 inflammasome is triggered in patients with CAPS and knock-in CAPS animal models that develop spontaneous systemic inflammatory responses from birth onwards in the absence of infection[21]. Recombinant expression of CAPS-associated *NLRP3* variants results in an open structure that favors inflammasome formation accompanied by a constitutive puncta distribution within the cell[22,23].

One challenge in studying gain-of-function *NLRP3* variants is the difficulty in distinguishing between the priming signal and the *NLRP3* expression per se. In this study, we use an inducible cellular system to express different gain-of-function *NLRP3* variants independently of the priming signal. We show that expression of CAPS-associated NLRP3 variants result in a constitutively active inflammasome that is regulated by ubiquitination. This results in a basal inflammatory programming and alter immunometabolism that limit IL-1β, but not IL-18 production. Therefore inflammatory flares in CAPS are controlled by glycolysis impairment, while maintaining a basal inflammatory program as a defence mechanism against pathogens.

## Results

### CAPS-associated NLRP3 variants result in a constitutively active inflammasome

We initially analysed blood samples from patients with CAPS carrying the germline pathogenic NLRP3 variants p.R260W, p.D303N, p.T348M and p.A439T[24]. Culturing whole blood from these patients without any stimulation resulted in an increased percentage of monocytes with ASC specks, used as a hallmark of inflammasome activation, when compared to healthy individuals (Fig. 1A). LPS stimulation further enhanced the percentage of ASC-specking monocytes in both groups, although this increase was less pronounced in healthy controls (Fig. 1A). In contrast, treatment with the NLRP3 inhibitor MCC950 (a.k.a. CRID3 or CP-456773) reduced the percentage of ASC-specking monocytes in both groups, with a marked reduction in CAPS patients (Supplementary Fig. 1). These results suggest that monocytes from CAPS patients exhibit constitutive NLRP3-inflammasome activation. Consistent with this, unstimulated peripheral blood mononuclear cells (PBMCs) from CAPS patients showed a constitutive release of the

pyroptotic marker galectin-3 and the cytokine IL-18 (Fig. 1B, C), with no detectable basal release of IL-1β or TNF-α (Fig. 1D,E). LPS stimulation led to an enhanced IL-1β release from the patients' PBMCs, but not from the healthy individuals (Fig. 1D), because is known that LPS is required for the transcriptional upregulation of IL-1β. We also found that LPS slightly, but significantly, increased the release of both IL-1β and IL-18 from the PBMCs of healthy individuals: IL-18 concentration increased from $0.78 \pm 0.13$ pg/ml at 6 h resting to $2.18 \pm 0.52$ pg/ml after 6 h of LPS treatment, with a $t$-test $p$-value of 0.0296; and IL-1β concentration increased from $0.65 \pm 0.28$ pg/ml at 4 h resting to $11.09 \pm 2.57$ pg/ml after 4 h of LPS treatment, with a $t$-test $p$-value of 0.0037 (Fig. 1C, D). However, this increase was less pronounced than that observed in the PBMCs of CAPS patients. This corresponds to the observed increase in ASC specks following LPS incubation in monocytes from healthy donors (Fig. 1A). We also observed a genotype-dependent differential release of galectin-3 and IL-1β, being higher in patients carrying the p.A439T variant compared to those carrying the remaining NLRP3 variants (Fig. 1B, D). Nevertheless, the observed increase in the percentage of ASC-specking monocytes was consistent across all analysed variants (Fig. 1A). This observation raises an intriguing hypothesis, while different NLRP3 variants may prompt constitutive inflammasome assembly, subsequent downstream signaling could be distinctly regulated. To investigate the effect of LPS stimulation on human monocyte activation, we analysed the GEO dataset GSE42606. The analysis revealed that LPS upregulates the expression of *IL1B*, *TNFA*, and *NLRP3* in healthy human PBMCs, while the expression of *IL18* and *LGALS3* remains unchanged (Fig. 1F). Contrarily, in active CAPS patients carrying the p.G569R NLRP3 variant, no discernible alteration in the expression patterns of *IL1B*, *IL18* and *NLRP3* in blood cells was observed, according to the analysis of GEO dataset GSE57253 (Fig. 1G). These findings collectively point towards the likelihood of an inherent constitutively NLRP3-inflammasome activation mechanism within CAPS.

### Expression levels of NLRP3 pathogenic variants condition the constitutive activation of the inflammasome

To investigate the activation of pathogenic NLRP3 variants independently of the NLRP3 inflammasome priming signal, we employed an inducible recombinant system in NLRP3-deficient immortalized mouse macrophages, where human *NLRP3* expression is regulated by doxycycline[22,25]. Inducing the expression of NLRP3 carrying the CAPS-associated variants p.R260W, p.T348M, and p.D303N, excluding the wild type, led to a constitutive processing of GSDMD (Fig. 2A). By increasing the concentration of doxycycline, we were able to enhance the expression level of mutant NLRP3, leading to increased GSDMD processing (Fig. 2A, Supplementary Fig. 2A). GSDMD processing was found to be dependent on NLRP3 activity, as evidenced by observing a reduction in GSDMD cleavage following treatment with the NLRP3 inhibitor MCC950 (Fig. 2A). Blocking GSDMD processing with MCC950 led to an increase in the expression of CAPS-associated NLRP3 variants in macrophages (Fig. 2A), probably due to the increase of *NLRP3* gene expression by non-dying cells. Moreover, by using different doxycycline doses to achieve similar expression levels of both the wild type and p.D303N NLRP3, GSDMD cleavage was exclusively observed in macrophages expressing the mutant NLRP3 variant (Supplementary Fig. 2B). Therefore, to compare the function of wild type and p.D303N NLRP3 at similar expression levels, these varying concentrations of doxycycline were used in subsequent experiments. Alongside GSDMD processing, NLRP3 p.D303N expression also led to caspase-1 activation (Fig. 2B), a heightened percentage of macrophages forming ASC oligomers (Fig. 2C, Supplementary Fig. 2C), and pyroptosis induction (Fig. 2D). We also observed that different CAPS-associated NLRP3 variants triggered the release of the constitutively expressed cytokine IL-18, which was inhibited by MCC950 (Fig. 2E). However, MCC950 presented a less potent inhibitory effect on the NLRP3 p.T348M

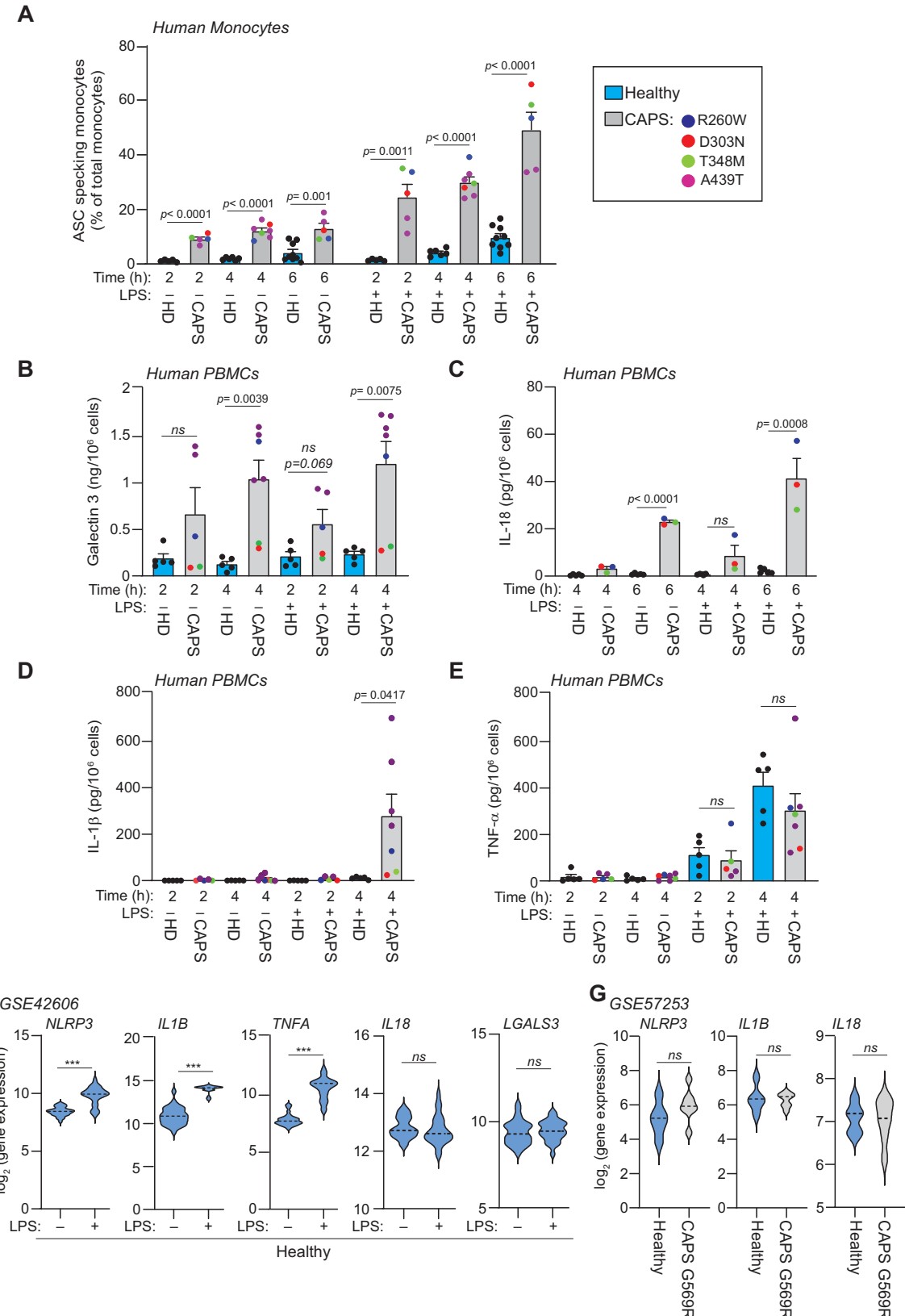

variant, affecting both GSDMD processing and IL-18 release (Fig. 2A, E). This observation aligns with a prior report indicating higher IC$_{50}$ for MCC950 in PBMCs from CAPS patients carrying the p.T348M variant compared to the p.A439V and p.E311K variants[9]. Furthermore, in recombinant systems, MCC950 was unable to completely eliminate cells with NLRP3 p.T348M-associated puncta[22]. These findings suggest

that therapy strategies involving MCC950 analogues might require dosage adjustments based on the specific *NLRP3* variant carried by patients.

The expression of the p.D303N NLRP3 variant was found to increase with the duration of doxycycline incubation (Fig. 3A). Interestingly, this increase was not observed during extended incubation

**Fig. 1 | Monocytes from CAPS patients show a constitutive inflammasome activation. A** Percentage of ASC specking monocytes from healthy donors (HD, blue bars, $n = 5$ for 2 h, $n = 6$ for 4 h, $n = 9$ for 6 h) and CAPS patients (gray bars, NLRP3 p.R260W, p.D303N, p.T348M and p.A439T, $n = 5$ for 2 and 6 h, $n = 7$ for 4 h) after whole blood treatment for different times with LPS (100 ng/ml). **B–E** ELISA for galectin 3 (**B**), IL-18 (**C**), IL-1β (**D**) and TNF-α (**E**) release from PBMCs treated as indicated in **A** ($n = 5$ HD, $n = 5$ CAPS patients for 2 h **B**, **D**, $n = 7$ CAPS patients for 2 h **B**, **D**, **E**, $n = 3$ CAPS patients for **C**). **F**, **G** Violin plot of *NLRP3*, *IL1B*, *TNFA*, *IL18* and *LGALS3* mRNA expression in PBMCs from healthy individuals treated or untreated with LPS (using dataset GSE42606) (**F**), or from blood cells of healthy individuals or

CAPS patients with p.G569R NLRP3 mutation during an inflammatory flare without LPS treatment (using dataset GSE57253) (**G**), the median is indicated by the middle line. For **A–E**, each dot corresponds to a different donor, and for CAPS, each color represents a different mutation; data is represented as mean ± SEM; *t*-test two-sided was used to compare between the group of healthy donors and CAPS patients in **A–E**; *t*-test two-sided and Wald test two-sided, both with Benjamini–Hochberg correction, were used to compare gene expression in **F**, **G**; significance levels are indicated as follows: \*\*\**p* < 0.0002; *ns* indicates no significant difference (*p* > 0.05). Source data are provided as a Source Data file.

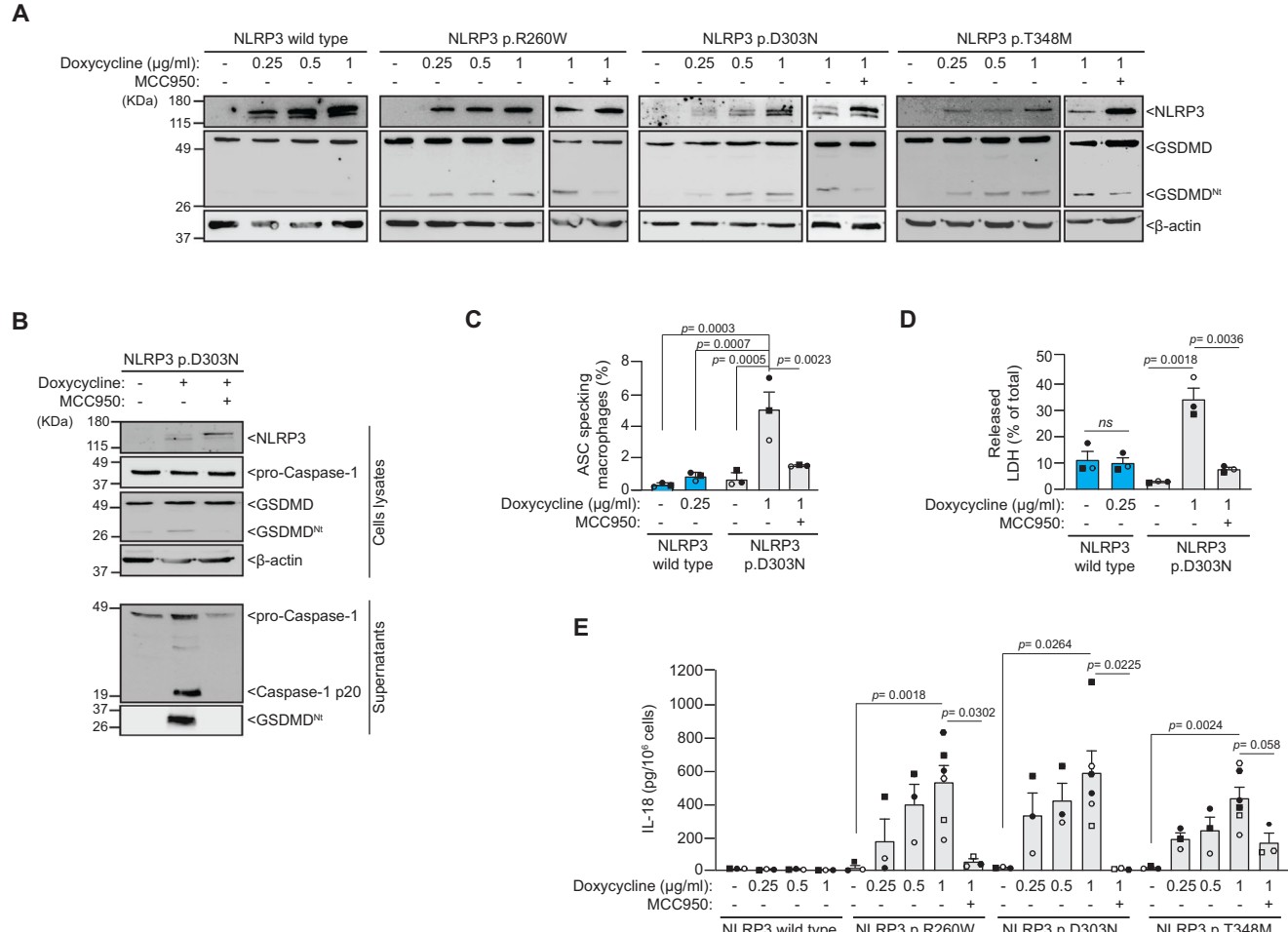

**Fig. 2 | Expression of CAPS-associated gain-of-function NLRP3 variants in macrophages results in constitutive active inflammasomes. A** Western blot for NLRP3, GSDMD, and β-actin in cell lysates from *Nlrp3*⁻/⁻ immortalized macrophages (iMos) treated for 16 h with or without doxycycline (0.25, 0.5 and 1 μg/ml) to induce the expression of the human wild type NLRP3 or p.D303N, p.R260W, and p.T348M variants in absence or presence of MCC950 (10 μM). **B** Western blot for NLRP3, caspase-1, GSDMD and β-actin in cell lysates and supernatants from iMos expressing the human NLRP3 p.D303N treated as indicated in **A** with doxycycline at 1 μg/ml. **C** Percentage of ASC specking iMos expressing the human wild type NLRP3 or

p.D303N variant treated as indicated in **A**, but with the annotated doxycycline concentration to obtain a similar expression of both NLRP3. **D** Percentage of extracellular LDH in iMos expressing human NLRP3 wild type or p.D303N variant treated as indicated in **C**. **E** ELISA for IL-18 release from iMos treated as indicated in **A**. Histograms in **C–E** present the mean ± SEM of $n = 3$ independent experiments (each one represented by a different symbol); Western blots are representative of $n = 3$ independent experiments; Ordinary one-way ANOVA test was used for **C** and **E**, *t*-test two-sided for **D**. Source data are provided as a Source Data file.

periods with doxycycline (16 h), likely due to the induction of pyroptosis. By contrast, blocking constitutive NLRP3 activation with MCC950 led to a sustained high expression of NLRP3 p.D303N, even during prolonged doxycycline exposure (Fig. 3A). These findings suggest that viable cells may promote the accumulation of inhibited NLRP3 p.D303N. Concurrently, augmented mutant NLRP3 expression triggered parallel GSDMD processing, LDH, and IL-18 release (Fig. 3A–C). To verify that the expression of CAPS-associated NLRP3 variants result in a constitutively active inflammasome, we conducted

a single-cell analysis using flow cytometry and Time of Flight for Inflammasome Evaluation (TOFIE) in HEK293T. This revealed an increased percentage of cells with ASC oligomers when the mutant p.D303N NLRP3 was equivalently expressed to the wild type NLRP3 (Fig. 3D, Supplementary Fig. 2D). In an attempt to mimic the typical heterozygous genotype of CAPS patients, we co-expressed both wild type and p.D303N NLRP3 in the same cells (Supplementary Fig. 2E). Data showed that the wild type NLRP3 induced a significant increase in cells with ASC oligomers at low p.D303N NLRP3 expression levels, but

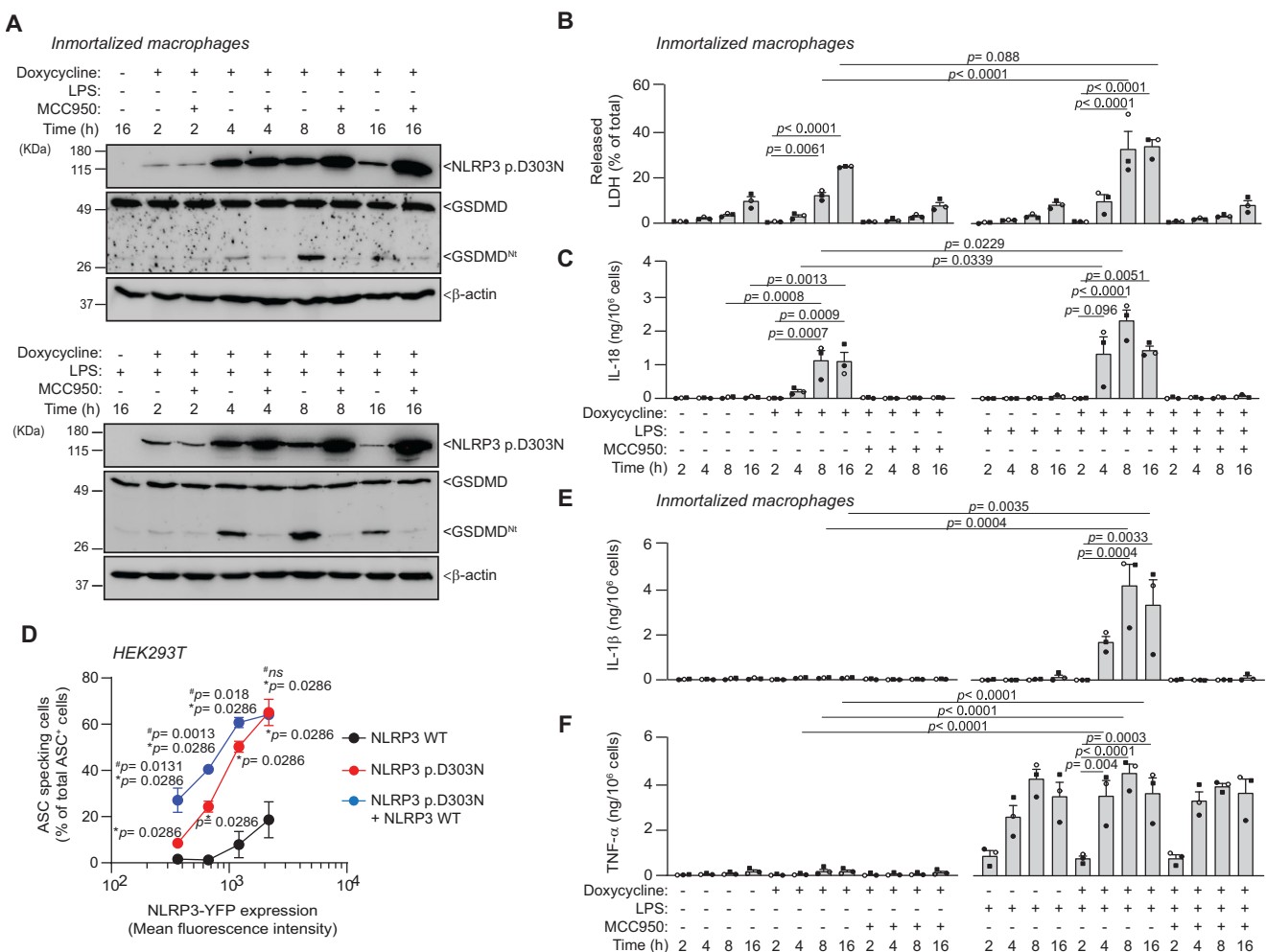

**Fig. 3 | Increased expression of NLRP3 p.D303N results in higher inflammasome activation. A** Western blot for NLRP3, GSDMD, and β-actin in cell lysates from *Nlrp3*[−/−] immortalized macrophages (iMos) treated for different times with or without 1 μg/ml doxycycline (2, 4, 8 or 16 h) to induce the expression of the human NLRP3 p.D303N, with or without LPS (100 ng/ml) and MCC950 (10 μM). **B, C** Extracellular LDH (**B**) and ELISA for IL-18 release (**C**) from iMos treated as described in **A**, each independent experiment is represented by a different symbol in the histograms. **D** Percentage of ASC specking HEK293T cells transfected with ASC-RFP and either NLRP3-YFP wild type (WT), the NLRP3 p.D303N-YFP variant, or a co-expression of both NLRP3 WT and p.D303N-YFP. The expression levels of NLRP3 were determined by an increase in the mean fluorescence intensity. **E, F** ELISA for IL-1β (**E**) and TNF-α (**F**) release from cells treated as indicated in **A**. Western blots are representative of *n* = 2 independent experiments; Graphics are representative of *n* = 3 independent experiments (each one represented by a different symbol in the histograms) and data are represented as mean ± SEM; Two-way ANOVA test was used for **B, C, E, F**; for **D** *t*-test two-sided to compare p.D303N and the co-trasfection was used and Mann−Whitney *U* test two-sided was used when the wild type was compared to the co-trasfection or the p.D303N alone, significance levels are indicated as follows: *ns* indicates no significant difference (*p* > 0.05). For **D**, *indicates comparison with the wild-type NLRP3, and [#]indicates comparison within the p.D303N NLRP3 and the co-expression. Source data are provided as a Source Data file.

not at high levels (Fig. 3D). This suggests that even low concentrations of mutant NLRP3 can enable the wild type to potentially intensify the basal autoactivation of the mutant NLRP3 inflammasome. This results in the oligomerization of ASC, caspase-1 activation, and GSDMD processing, all occurring independently of any priming signal. These results align with previous observation that CAPS-associated NLRP3 variants inherently display an open active structure and a constitutive puncta distribution within the cell without external stimulation[22].

**NF-κB induction modulates the secretome of pathogenic NLRP3 variants**

Consistent with the results obtained from monocytes analysis of CAPS patients as shown in Fig. 1, the expression of pathogenic NLRP3 variants in immortalized macrophages did not trigger a significant release of IL-1β. As expected, IL-1β release was strongly induced following LPS treatment (Fig. 3E). LPS also promoted the release of TNF-α, which was independent of the expression of p.D303N NLRP3 variant, as

evidenced by the fact that MCC950 did not modify TNF-α release but completely inhibited IL-1β release (Fig. 3E, F). Interestingly, LPS stimulation also enhanced IL-18 release and GSDMD cleavage when the p.D303N NLRP3 variant was expressed (Fig. 3A, C). Aligning with this observation, other NF-κB activators, such as palmitate, S100A9 or IL-6 acted as inductors of NLRP3-dependent IL-1β release when pathogenic NLRP3 variants p.R260W, p.D303N and p.T348M were expressed, which was not the case with the expression of the wild type allele (Fig. 4A, Supplementary Fig. 3A–C). For control, wild type NLRP3 expression induced IL-1β release following LPS and nigericin treatment (Supplementary Fig. 3D). Additionally, palmitate, S100A9 and IL-6 induced TNF-α release regardless of the expressed NLRP3 variants (Fig. 4B). These compounds were all able to activate NF-κB in a macrophage reporter system (Fig. 4C). As expected, MCC950 inhibited IL-1β release from macrophages expressing the p.D303N NLRP3 variant, but did not affect TNF-α production (Fig. 4A, B). Furthermore, the TLR agonists LPS and Pam3-CSK₄ also triggered IL-1β release from

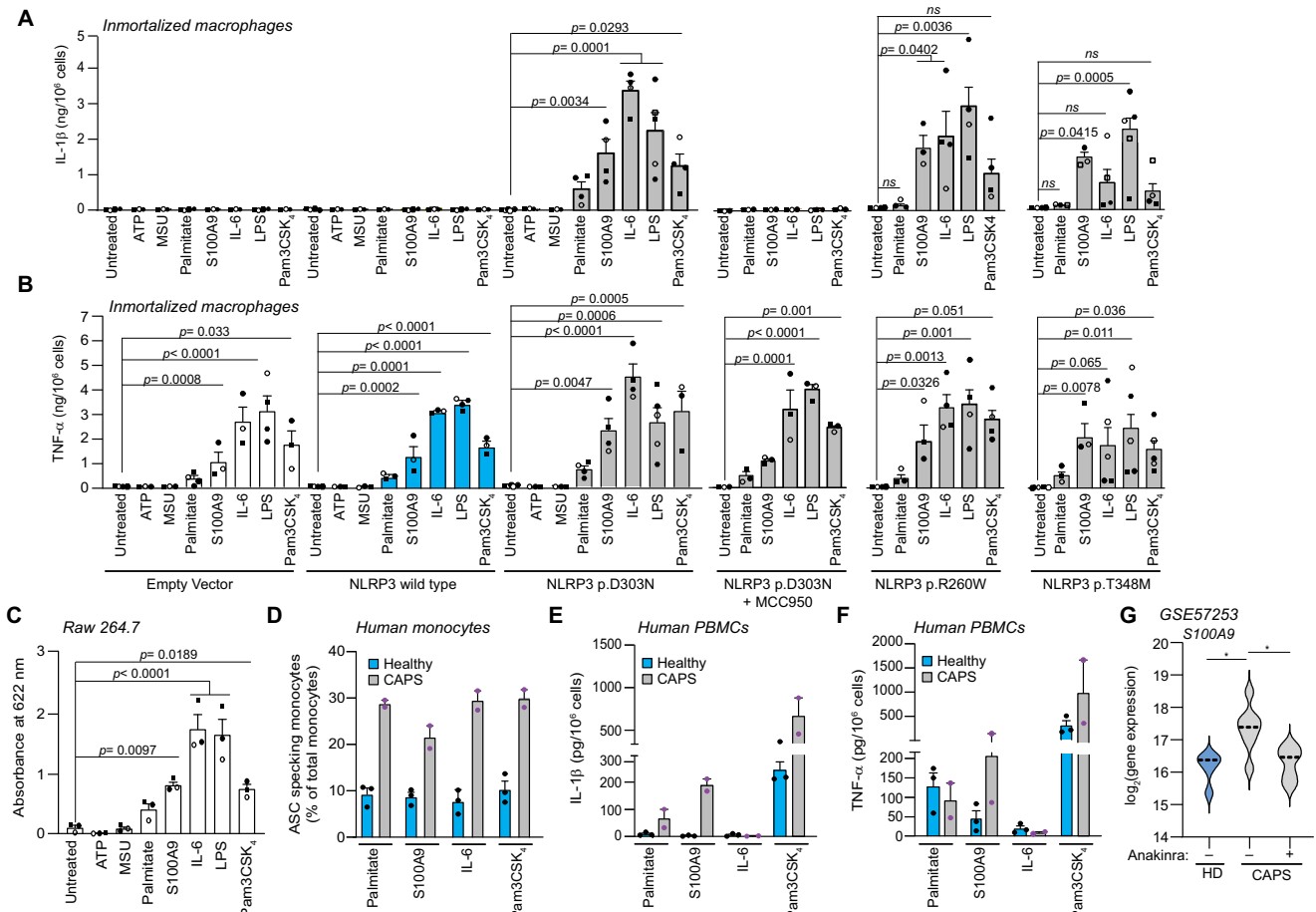

**Fig. 4 | PAMPs and DAMPs are required to induce NF-κB–dependent IL-1β release from macrophages expressing pathogenic NLRP3 variants. A, B** ELISA for IL-1β (**A**) and TNF-α (**B**) released from *Nlrp3⁻/⁻* immortalized macrophages (iMos) expressing the human wild type NLRP3 or the p.D303N, p.R260W, and p.T348M variants induced after 16 h treatment with doxycycline (1 μg/ml) and ATP (5 mM), MSU crystals (200 μg/ml), palmitate (1 mM), S100A9 (0.5 μg/ml), IL-6 (0.5 μg/ml), LPS (100 ng/ml) or Pam3-CSK₄ (1 μg/ml), together with or without MCC950 (10 μM). **C** Colorimetric quantification of the levels of secreted alkaline phosphatase (SEAP) from NF-κB reporter RAW 264.7 macrophages treated for 4 h with ATP (5 mM), MSU crystals (200 μg/ml), palmitate (1 mM), S100A9 (0.5 μg/ml), IL-6 (0.5 μg/ml), LPS (0.1 μg/ml) or Pam3-CSK₄ (1 μg/ml). **D** Percentage of ASC specking monocytes from p.A439T CAPS patients (gray bars) and healthy donors (blue bars) after whole blood treated during 6 h with Pam3CSK₄ (1 μg/ml), IL-6 (0.5 μg/ml), S100A9 (0.5 μg/ml) or palmitate (1 mM). **E, F** ELISA for IL-1β (**E**) and

TNF-α (**F**) release from healthy donors and p.A439T CAPS patients PBMCs treated as indicated in **D**. **G** Violin plot of *S100A9* mRNA expression from blood cells of healthy individuals or CAPS patients with the p.G569R NLRP3 mutation during active disease or treated with anakinra (using dataset GSE57253). The median indicated by the middle line. For exact *n* numbers of **A, B** see 'Statistics and reproducibility' section in Methods; for **C** *n* = 3 independent experiments (each one represented by a different symbol); for **D–F**, each dot corresponds to a different donor (*n* = 3 healthy donors, *n* = 2 CAPS); and for **G** *n* = 5 healthy donors, *n* = 7 CAPS patients during active disease and the same *n* = 7 patients treated with anakinra; For **A–F** data are represented as mean ± SEM; Ordinary one-way ANOVA test for **A–C**, except for p.T348M mutant in **B** one-way Kruskal–Wallis; Wald test two-sided with Benjamini–Hochberg correction for **G**; significance levels are indicated as follows: *p < 0.05; *ns* indicates no significant difference (*p* > 0.05). Source data are provided as a Source Data file.

macrophages expressing the pathogenic NLRP3 variants (Fig. 4A), and were also capable of inducing NF-κB and TNF-α release (Fig. 4B, C). Notably, the p.T348M NLRP3 variant exhibited a subdued IL-1β release compared to the p.R260W and p.D303N NLRP3 variants (Fig. 4A). Both palmitate and IL-6 were more effective in inducing IL-1β release when the p.D303N NLRP3 variant was expressed (Fig. 4A).

The stimulation of whole blood from CAPS patients with palmitate, S100A9, IL-6 and Pam3-CSK₄ led to an increase in the percentage of ASC-specking monocytes. However, this increase was less pronounced in equally treated monocytes from healthy individuals (Fig. 4D). IL-1β release from CAPS PBMCs was triggered after stimulation with palmitate, S100A9 and Pam3-CSK₄, whereas unstimulated cells or those treated with recombinant IL-6 did not elicit this response (Fig. 4E). Serving as control, TNF-α release was also induced in PBMCs from both CAPS patients and healthy individuals treated with palmitate, S100A9 and Pam3-CSK₄, but not by IL-6 (Fig. 4F). This absence of response may stem from IL-6 inability to activate the NF-κB pathway in

samples from CAPS patients. Further examination of the GEO dataset GSE57253 revealed increased *S100A9* gene expression in blood cells from active CAPS patients when compared to healthy donors (Fig. 4G). Remarkably, anakinra treatment in CAPS patients reduced *S100A9* gene expression (Fig. 4G). This suggests that alarmins, such as S100A9, could be involved in CAPS flares by enhancing the activity of the mutant NLRP3 inflammasome.

Triggers of the canonical wild type NLRP3 inflammasome, such as extracellular adenosine 5'-triphosphate (ATP) or monosodium urate (MSU) crystals failed to induce IL-1β release from immortalized macrophages expressing the pathogenic NLRP3 variants (Fig. 4A), as these compounds were unable to activate NF-κB (Fig. 4C). We observed a dose-dependent release of IL-1β from macrophages expressing the p.D303N NLRP3 variant upon exposure to increasing concentrations of palmitate, S100A9, IL-6, LPS and Pam3-CSK₄ (Supplementary Fig. 3A), similar to the TNF-α release pattern (Supplementary Fig. 3A). However, there was a noticeable delay in the release kinetics of IL-1β in

comparison to TNF-α release (Supplementary Fig. 3B). This underscores the dependency of IL-1β release in pathogenic NLRP3 variants on two distinct processes: initially the by NF-κB pathway stimulates the expression of pro-IL-1β (Supplementary Fig. 3C), which is then followed by its cleavage and release due to the constitutively active pathogenic NLRP3 inflammasome.

Given our observation of increased GSDMD cleavage and IL-18 release after LPS stimulation (Fig. 3A, B), we aimed to investigate the possibility of NF-κB activation modulating the constitutively active p.D303N NLRP3 inflammasome. We found a slight increased processing of caspase-1 p20 and GSDMD processing following NF-κB induction with palmitate, S100A9, IL-6, LPS and Pam3-CSK₄ (Fig. 5A). Consistently, treatment of macrophages expressing the p.D303N NLRP3 variant with these compounds resulted in a significant increase in ASC specking macrophages (Fig. 5B), as well as IL-18 and IL-1α release (Fig. 5C, D), P2X7 receptor shedding (Fig. 5E), and pyroptosis (Fig. 5F). As expected, MCC950 was able to reduce all these effects (Fig. 5C–F). These data suggest that while IL-1β release was strongly dependent on NF-κB induction, the constitutive activity of the mutant NLRP3 inflammasome was slightly modulated by NF-κB at a constant NLRP3 expression.

NF-κB activation from macrophages expressing the p.D303N NLRP3 variant was also crucial for the inflammasome-dependent release of HMGB1 and cystatin B (Supplementary Fig. 4A,B), with palmitate being the weakest inducer (Supplementary Fig. 4A,B) and the weakest NF-κB activator (Fig. 4C). In contrast, the pathogenic p.D303N NLRP3 variant was not implicated in the release of cathepsin B, CD14, CD206 or annexin A1 (Supplementary Fig. 4C–F), which were previously identified to be released upon canonical inflammasome activation by extracellular ATP[26].

## Ubiquitin regulates the basal activity of mutant NLRP3 inflammasome

Deubiquitinases (DUBs) are known to play a role in the priming and activation of the canonical wild type NLRP3 inflammasome[27]. Consequently, we next aimed to investigate whether DUBs could also be regulating the activity of the mutant NLRP3 inflammasome. The application of broad-spectrum DUBs inhibitors, G5 and PR-619, in macrophages expressing the p.D303N NLRP3 variant resulted in a reduction of GSDMD processing (Fig. 6A) and IL-18 release (Fig. 6B). Similarly, the administration of the selective USP14 and UCHL5 inhibitor, b-AP15, also led to a decrease in GSDMD processing and IL-18 release (Fig. 6A,B), thus suggesting that deubiquitination of the mutant NLRP3 may favor the constitutive activation of the CAPS-related inflammasome. Moreover, the different DUB inhibitors were also capable of reducing GSDMD processing following LPS treatment of macrophages expressing the p.D303N NLRP3 variant (Fig. 6C), without impacting LPS-induced NF-κB activation in the NF-κB reporter Raw264.7 cells (Fig. 6D) or TNF-α release (Fig. 6E). In the presence of LPS, the different DUB inhibitors further decreased the release of inflammasome-dependent cytokines IL-18 and IL-1β (Fig. 6F, G). Notably, both G5 and b-AP15 exhibited a mild reductive effect on the percentage of ASC-specking monocytes in untreated blood samples from CAPS patients, with this decline being more pronounced in LPS-treated blood samples (Fig. 6H). Although NF-κB activation was not affected by the DUB inhibitors (Fig. 6D), we discerned that all employed DUB inhibitors (G5, PR-619 and b-AP15) induced a significant decrease in LPS-induced *Il1b* and *Il18* gene expression, without affecting the expression of *Tnfa*, *Pycard*, *Casp1*, *Gsdmd*, *Nek7* or *Nlrp3* (Supplementary Fig. 5). This suggests that DUBs might be modulating mutant NLRP3 inflammasome activity beyond merely influencing *Il1b* and *Il18* gene expression. Collectively, our data indicates that DUBs control the constitutive activation of the CAPS-associated NLRP3-inflammasome, as observed in both circulating monocytes from CAPS patients and immortalized macrophages recombinantly expressing pathogenic NLRP3 variants.

## Constitutive active mutant NLRP3 inflammasome induces metabolic reprogramming

Given that an inflammatory state can rapidly shift oxidative metabolism towards glycolysis[28,29], we next aimed to evaluate whether the constitutive active mutant NLRP3-inflammasome could impact the metabolism of myeloid cells. For this purpose, monocytes isolated from healthy individuals or CAPS patients carrying the pathogenic p.A439T NLRP3 variant, either unstimulated or treated with LPS, underwent untargeted metabolomics analysis. The pre-processing of the entire metabolite dataset resulted in a data matrix comprising 203 metabolites in negative polarity and 925 in positive polarity, from which they were reduced to 603 metabolites in positive polarity after removal metabolites with high variability among replicates (Supplementary Data 1). To efficiently represent the variability of these metabolites among monocytes of healthy donors and CAPS patients while preserving trends and patterns, we employed a principal component analysis (PCA) model on the final merged data matrix (Fig. 7A). The calculated PCA model was built based on 16 samples and three components, with the first two principal components (PC1 and PC2) accounting for 25.2% and 9.6% of the overall variability, respectively. PCA model results illustrated variations based on the total covariance among the study cohorts. We observed that the metabolomic variation of the untreated monocytes from CAPS patients clustered more closely with the LPS-treated monocytes from healthy individuals (Fig. 7A), suggesting an inherent pro-inflammatory profile of CAPS monocytes. In fact, both unstimulated monocytes from CAPS patients and those from healthy donors treated with LPS were mainly explained by component 1, indicating that the differences evidenced in sample metabolomes might be influenced by these variables. This observation was further confirmed, as 423 metabolites displayed differential presence in monocytes from healthy individuals and CAPS patients (Fig. 7B). Of these, 46 were found to be upregulated in LPS-treated monocytes from healthy donors. Interestingly, these same metabolites were also found to be upregulated in both untreated and LPS-treated monocytes from CAPS patients, when compared to untreated monocytes from healthy donors (Fig. 7B). Subsequently, 121 metabolites were observed to be upregulated, while 256 metabolites were found to be downregulated in monocytes from CAPS when compared to monocytes from healthy individuals (Fig. 7B). Volcano plots further revealed significant differences in the metabolites when contrasting monocytes from CAPS patients with those from healthy donors, both under resting conditions and post-LPS treatment (Fig. 7C). Aligning with the pro-inflammatory metabolomic landscape of the monocytes from CAPS patients, differences in metabolites were less pronounced when comparing untreated monocytes from CAPS patients with LPS-treated monocytes from healthy individuals (Supplementary Fig. 6A). The untargeted metabolomics analysis conducted was not sufficiently conclusive in identifying the metabolites that varied between monocytes from healthy individuals and those from CAPS patients, due to limited sample availability for the targeted MS/MS analysis required for metabolite confirmation through spectra fragmentation patterns. However, we managed to tentatively identify 14 metabolites, with an error rate ranging from −2.68 to 8.52 ppm, which exhibited differential presence in monocytes of the two groups (Supplementary Table 2, Supplementary Fig. 6B). These included lipids and lipid-like molecules, as well as amino acids and their synthesis intermediates, most of which were derived from glycolysis intermediates (Supplementary Table 2, Supplementary Fig. 6B).

To further validate changes in metabolites associated with the auto-active mutant NLRP3 inflammasome, we conducted an untargeted metabolomics analysis in immortalized macrophages expressing the pathogenic p.D303N NLRP3 variant. The pre-processing

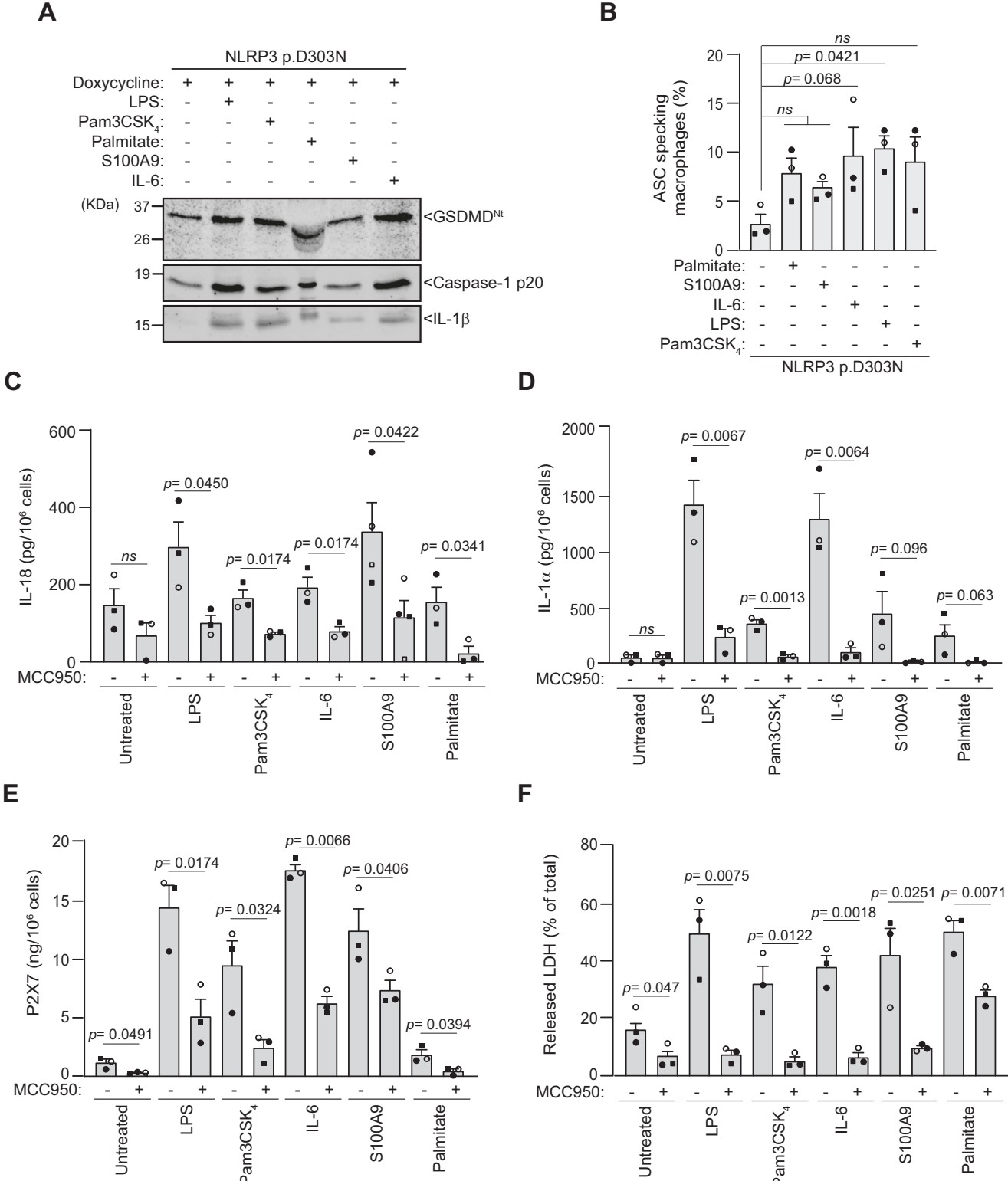

**Fig. 5 | NF-κB induction enhances pathogenic NLRP3 inflammasome activation.**
**A** Western blot for GSDMD, caspase-1 and IL-1β in supernatants from *Nlrp3*[-/-] immortalized macrophages (iMos) expressing human NLRP3 p.D303N mutant induced after 16 h treatment with doxycycline (1 μg/ml) and palmitate (1 mM), recombinant S100A9 (0.5 μg/ml), recombinant IL-6 (0.5 μg/ml), LPS (0.1 μg/ml) or Pam3-CSK₄ (1 μg/ml). **B** Percentage of ASC specking iMos expressing human NLRP3 p.D303N mutant treated as described in **A**. **C–E** ELISA for the release of IL-18 (**C**), IL-1α (**D**), soluble P2X7 receptor (**E**) from iMos expressing human NLRP3 p.D303N

mutant treated as described in **A**, but with or without MCC950 (10 μM).
**F** Percentage of extracellular LDH in iMos expressing human NLRP3 p.D303N mutant treated as indicated in **C**. Western blots are representative of *n* = 3 independent experiments. For **B**–**F** *n* = 3 independent experiments (each one represented by a different symbol), except for **C** where S100A9 treatment is *n* = 4, and data is presented as mean ± SEM; Ordinary one-way ANOVA test was used for **B**, and *t*-test two-sided in **C**–**F**; *ns* indicates no significant difference (*p* > 0.05). Source data are provided as a Source Data file.

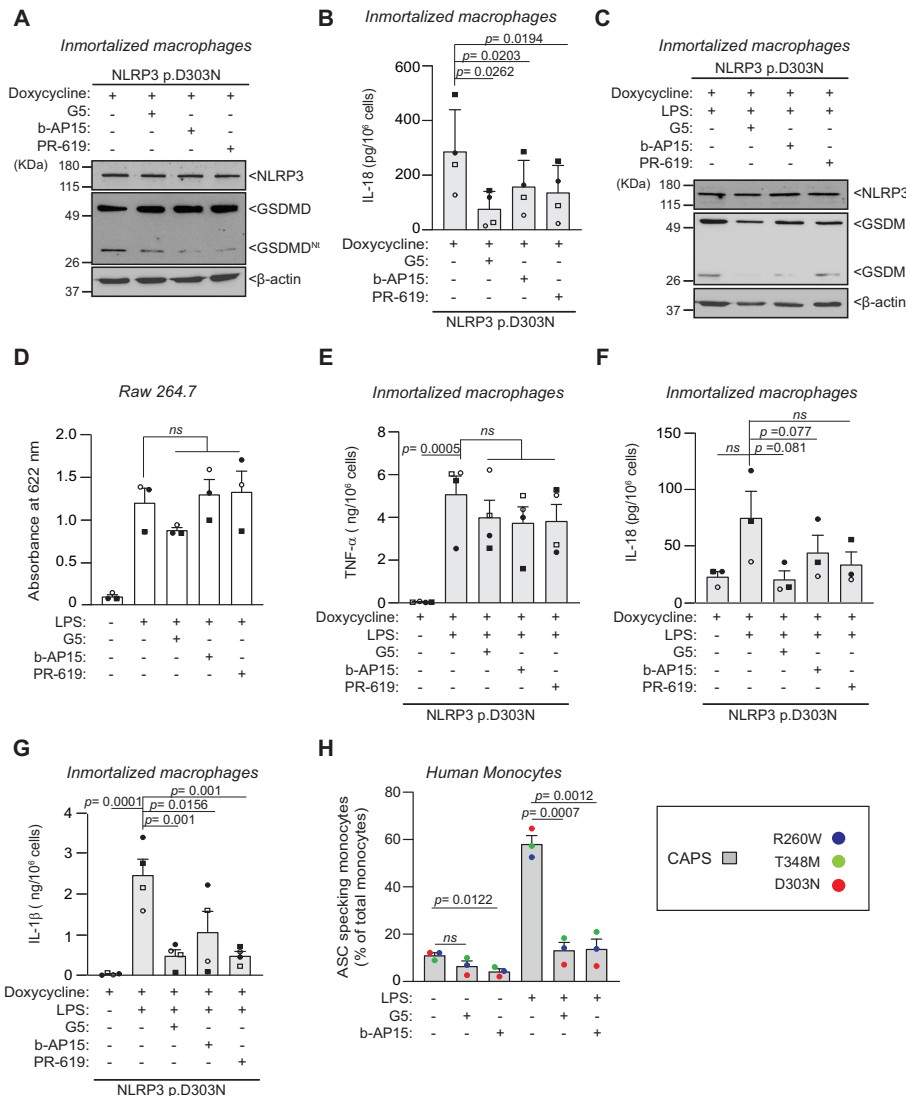

**Fig. 6 | Deubiquitinases control pathogenic NLRP3 inflammasome activity.**
**A** Western blot for NLRP3, GSDMD and β-actin in cell lysates from *Nlrp3⁻/⁻*
immortalized macrophages (iMos) expressing the human NLRP3 p.D303N mutant
induced after 16 h treatment with doxycycline (1 μg/ml), in the absence or presence
of G5 (5 μM), b-AP15 (5 μM) or PR-619 (10 μM). **B** ELISA for IL-18 release from iMos
treated as described in **A**. **C** Western blot for NLRP3, GSDMD and β-actin in cell
lysates from *Nlrp3⁻/⁻* iMos expressing human NLRP3 p.D303N mutant induced after
16 h treatment with doxycycline (1 μg/ml), and then treated for 6 h with LPS
(0.1 μg/ml) in the absence or presence of G5 (5 μM), b-AP15 (5 μM) or PR-619 (10 μM).
**D** Colorimetric quantification of the levels of secreted alkaline phosphatase (SEAP)
from Raw264.7 macrophages expressing SEAP reporter gen under the control of
the NF-κB promoter treated for 6 h with LPS (0.1 μg/ml) in the absence or presence

of G5 (5 μM) or b-AP15 (5 μM) or PR619 (10 μM). **E–G** ELISA for TNF-α (**E**), IL-18 (**F**)
and IL-1β (**G**) release from iMos treated as described in **C**. **H** Percentage of ASC
specking monocytes from CAPS patients after whole blood treated for 6 h with LPS
(0.1 μg/ml) in the absence or presence of G5 (5 μM) or b-AP15 (5 μM). Blots are
representative of $n = 3$ independent experiments. For **D**, **F** $n = 3$ and for **B**, **E**, **G** $n = 4$
independent experiments (each one is represented by a different symbol); For
**H** each dot corresponds to a different CAPS donor, $n = 3$ patients and each color
represents a different mutation as indicated; Histogram data is represented as
mean ± SEM; Ordinary one-way ANOVA test was used for **D**, **E**, **G** and *t*-test two-
sided was used for **B**, **F**, **H**; *ns* indicates no significant difference ($p > 0.05$). Source
data are provided as a Source Data file.

operations from the full metabolites data set yielded a data matrix
based on 243 metabolites and 636 metabolites in negative and positive
polarities respectively (Supplementary Data 2), with 166 metabolites
upregulated in immortalized macrophages expressing mutant NLRP3.
The calculated PCA model was constructed based on 12 samples and
three components. The PC1 and PC2 accounted for 29.5% and 15.9% of
the total variability respectively, and indicated that the macrophages
expressing NLRP3 p.D303N variant formed a group that was similar to
the LPS-treated macrophages that were not expressing NLRP3 (Sup-
plementary Fig. 6C). A tentative identification of 17 metabolites, with
an error rate ranging from −2.69 to 8.87 ppm, were also found to be
enriched in lipids, amino acid and their derivative/intermediate

metabolites (Supplementary Table 3, Supplementary Fig. 6B). This
confirms that the expression of pathogenic NLRP3 variants affects
distinct metabolic pathway in CAPS monocytes.

**Glycolysis is impaired by the active mutant NLRP3
inflammasome**
We then proceeded to examine the expression of metabolic-associated
gene expression in two independent GEO datasets (GSE57253 and
GSE17732) that included blood samples obtained from CAPS patients
during an active disease periods[30,31]. Interestingly, our analysis
revealed a marked differential expression of genes associated with
glycolysis and the pentose phosphate pathway in blood cells of active

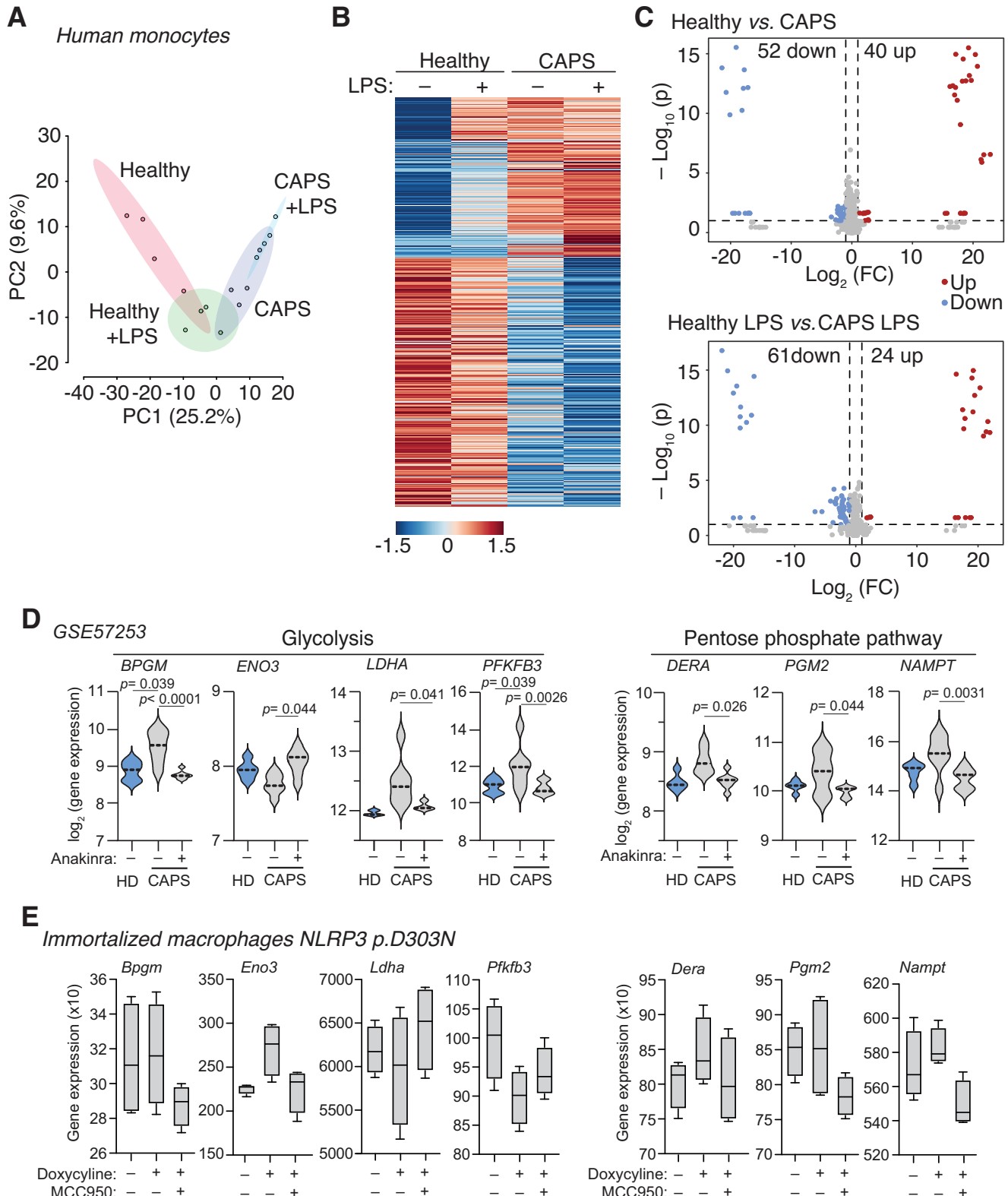

CAPS patients. Moreover, upon patients treatment with IL-1 inhibitor anakinra, the expression of these genes appear to normalize (Fig. 7D, Supplementary Fig. 7A). Genes associated with pentose phosphate (*DERA, PGM2, NAMPT*) and glycolysis (*BPGM, LDHA, PFKFB3*) revealed elevated expression in blood cells from active CAPS patients, and a decrease following anakinra treatment (Fig. 7D, Supplementary Fig. 7A). In contrast, the expression of the glycolytic enzyme encoded by the *ENO3* gene tended to decrease during active CAPS and was

restored following anakinra treatment (Fig. 7D, Supplementary Fig. 7A). A similar gene expression assessment in immortalized macrophages expressing the mutant p.D303N NLRP3 induced by doxycycline, revealed a predominant upregulation trend for genes associated with the pentose phosphate pathway (Fig. 7E), contrasting with a downregulation for the glycolytic-related genes (Fig. 7E). This discrepancy might be because in the case of CAPS patients, the blood samples for these studies were drawn during an inflammatory flare,

**Fig. 7 | Monocytes from CAPS patients present an altered metabolism.**
**A** Principal component analysis (PCA) model plot of metabolomic profiles of monocytes from healthy subjects ($n = 4$) and CAPS ($n = 4$) incubated or not for 2 h with LPS (500 ng/ml). **B** Metabolomic profiles of monocytes from healthy subjects (average of $n = 4$) and CAPS (average of $n = 4$) incubated or not for 2 h with LPS (500 ng/ml). **C** Volcano plots of blood monocytes from healthy subjects ($n = 4$) and CAPS patients ($n = 4$), with expression (log$_2$ values) plotted against the adjusted *P* value for the difference metabolite abundance, *t*-test two-sided. Significantly upregulated (red) or downregulated (blue) metabolites one-fold or more in monocytes from CAPS patients relative to that in monocytes from healthy subjects. **D** Violin plot of mRNA expression from genes of the glycolysis (*BPGM, ENO3, LDHA, PFKFB3*) and pentose phosphate pathway (*DERA, PGM2, NAMPT*) from blood cells of healthy donors (HD) or CAPS patients with p.G569R NLRP3 mutation during active

disease or after treatment with anakinra (using dataset GSE57253). Middle dotted line depicts the median and graphic represents data from $n = 5$ healthy donors, $n = 7$ CAPS patients during active disease and the same $n = 7$ CAPS patients CAPS treated with anakinra; Wald test two-sided, with Benjamini–Hochberg correction, were used to compare gene expression. **E** Boxplot of mRNA expression from genes of the glycolysis (*Bpgm, Eno3, Ldha, Pfkfb3*) and pentose phosphate pathway (*Dera, Pgm2, Nampt*) from *Nlrp3*$^{-/-}$ immortalized macrophages treated for 16 h with or without doxycycline (1 µg/ml) to induce the expression of human NLRP3 p.D303N variant in absence/presence of MCC950 (10 µM). Middle line depicts the mean, error bars represent minimum and maximum, and bounds of box represent the 25$^{th}$ to 75$^{th}$ percentile respectively; graphic represents data from $n = 4$ independent experiments. Source data are provided as a Source Data file.

whereas immortalized macrophages were unprimed and had not encountered any pro-inflammatory stimuli. Indeed, when CAPS patients were treated with anakinra, the same induced downregulation in glycolytic gene expression as was found, mirroring the findings seen when the p.D303N NLRP3 was expressed in immortalized macrophages (Fig. 7D, E). A full analysis of RNA sequencing of immortalized macrophages expressing the p.D303N NLRP3 variant showed that most of the glycolytic genes were down-regulated and their expression was recovered with MCC950 treatment (Fig. 8A, B). Expression of the wild type NLRP3 in immortalized macrophages did not significantly influence the expression of glycolytic genes (Fig. 8B). On the contrary, genes associated with the pentose phosphate pathway clustered with those that were upregulated upon expression of p.D303N NLRP3 (Fig. 8A, Supplementary Fig. 7B). This group also included genes related to the inflammatory response, apoptosis, lipid and amino acid metabolism, immune responses to virus, oxidative stress and IL-17 (Fig. 8A,C, Supplementary Table 4). This underlines that CAPS exhibit changes in lipids, amino acids and associated intermediates, along with a basal inflammatory state, potentially enhancing the ability to counteract pathogens (since cellular pathways related to virus response were significantly enriched). Among inflammatory genes, we detected upregulation of the nuclear receptors 4A1 and 4A2, key for the activation of an inflammatory program in macrophages. When analysing biological processes downregulated upon p.D303N NLRP3 expression and restored with MCC950 treatment, we identified pathways related to cellular homeostatic functions and tissue regeneration, including transcription, cell proliferation, cytoskeleton organization, angiogenesis regulation, cell migration or ERK cascades (Fig. 8C, Supplementary Table 5). Importantly, genes associated to hypoxic responses and the metabolism of lipids and carbohydrates were down regulated upon the expression of the mutant p.D303N NLRP3 (Fig. 8C, Supplementary Table 5).

To further investigate whether alterations in gene expression led to functional changes in macrophages expressing either mutant or wild-type NLRP3, we examined glycolysis functionality. To achieve this, we evaluated glycolysis rate by measuring the acidification of extracellular media produced by mitochondria in response to proton efflux and measured the oxygen consumption rate of these macrophages. Our results indicated that macrophages expressing the mutant NLRP3 variant displayed decreased basal and compensatory glycolysis (following mitochondrial respiration inhibition), regardless of whether they were treated with LPS or not (Fig. 9A, B). Administrating MCC950 restored the glycolytic activity of the macrophages expressing the mutant p.D303N NLRP3 (Fig. 9A,B), simultaneously to an upregulation in glycolytic-related gene expression (Fig. 8B). Neither doxycycline nor MCC950 had any effect in the glycolysis of *Nlrp3*$^{-/-}$ macrophages infected with the empty vector virus, or in macrophages expressing wild type NLRP3 (Fig. 9B). The reduction in glycolysis was a result of the p.D303N NLRP3 inflammasome activity, as it was reversed by MCC950 (Fig. 9A, B). However, this effect was not due to IL-1 downstream signaling, as the glycolytic rate remained unaffected when

incubated with recombinant IL-1Ra (Supplementary Fig. 7C). Furthermore, there was a noticeable decrease in the production of lactate, pyruvate, and glycolytic ATP when the mutant NLRP3 was expressed, and this reduction was mitigated upon administering MCC950 (Fig. 9C, D, Supplementary Fig. 8A). Even though the expression of mutant NLRP3 also influenced mitochondrial ATP production (Fig. 9), it did not alter mitochondrial basal or maximal respiration in resting macrophages, while slightly decreasing in LPS-treated macrophages (Fig. 9E, Supplementary Fig. 8B). Similarly, metabolites from the tricarboxylic acid cycle such as citrate, malate, fumarate, succinate, and ketoglutarate, along with their associated genes, remained unchanged after the expression of the p.D303N NLRP3 variant (Supplementary Fig. 8C, D).

### Glycolysis impairment due to constitutive active mutant NLRP3 inflammasome limit basal IL-1β production

To gain insights into the functional consequences of the decreased of glycolysis associated with pathogenic NLRP3 variants, we supplemented immortalized macrophages cultures expressing mutant NLRP3 variant with pyruvate, the end-product of glycolysis. Pyruvate was able to increase the release of IL-1β, but not of IL-18, in unprimed macrophages expressing the p.D303N NLRP3 variant (Fig. 10A). Pyruvate did not affected IL-1β release when the wild type NLRP3 was expressed in the macrophages (Fig. 10A). In fact, when the p.D303N variant was expressed in macrophages, there was a subtle elevation in IL-1β gene expression, an effect not seen with the wild-type NLRP3 (Fig. 10B). Contrarily, when glycolysis was impaired by blocking hexokinase activity with 2-deoxy-D-glucose (2DG) (Supplementary Fig. 8E), we observed a decrease in IL-1β release, but not in IL-18 (Fig. 10C). As expected, 2DG also blocked IL-1β release upon canonical activation of the wild type NLRP3 (Supplementary Fig. 8F). In summary, our findings show that NLRP3 harboring CAPS-associated pathogenic variants produces basal active inflammasomes even without cell priming. This spontaneous activation leads to the constitutive production of IL-18 and changes in the immunometabolism, marked by reduced glycolysis which in turn lessens IL-1β production, limiting inflammatory flares but producing a positive basal protection against pathogens.

### Discussion

Our study reveals that pathogenic variants in *NLRP3* associated with CAPS can constitutively activate NLRP3 inflammasomes even in the absence of triggers or priming signals. The expression level of this pathogenic NLRP3 variants directly associates with increased inflammasome activity (ASC oligomer formation, caspase-1 activation, GSDMD processing, IL-18 release and pyroptosis), as well as heightened basal inflammatory gene expression (i.e., upregulation of nuclear receptor 4 A). We discovered that triggers activating NF-κB pathway enhance the activation of the inflammasome carrying pathogenic NLRP3 variants and broaden the range of molecules released from macrophages. A notable outcome is a significant

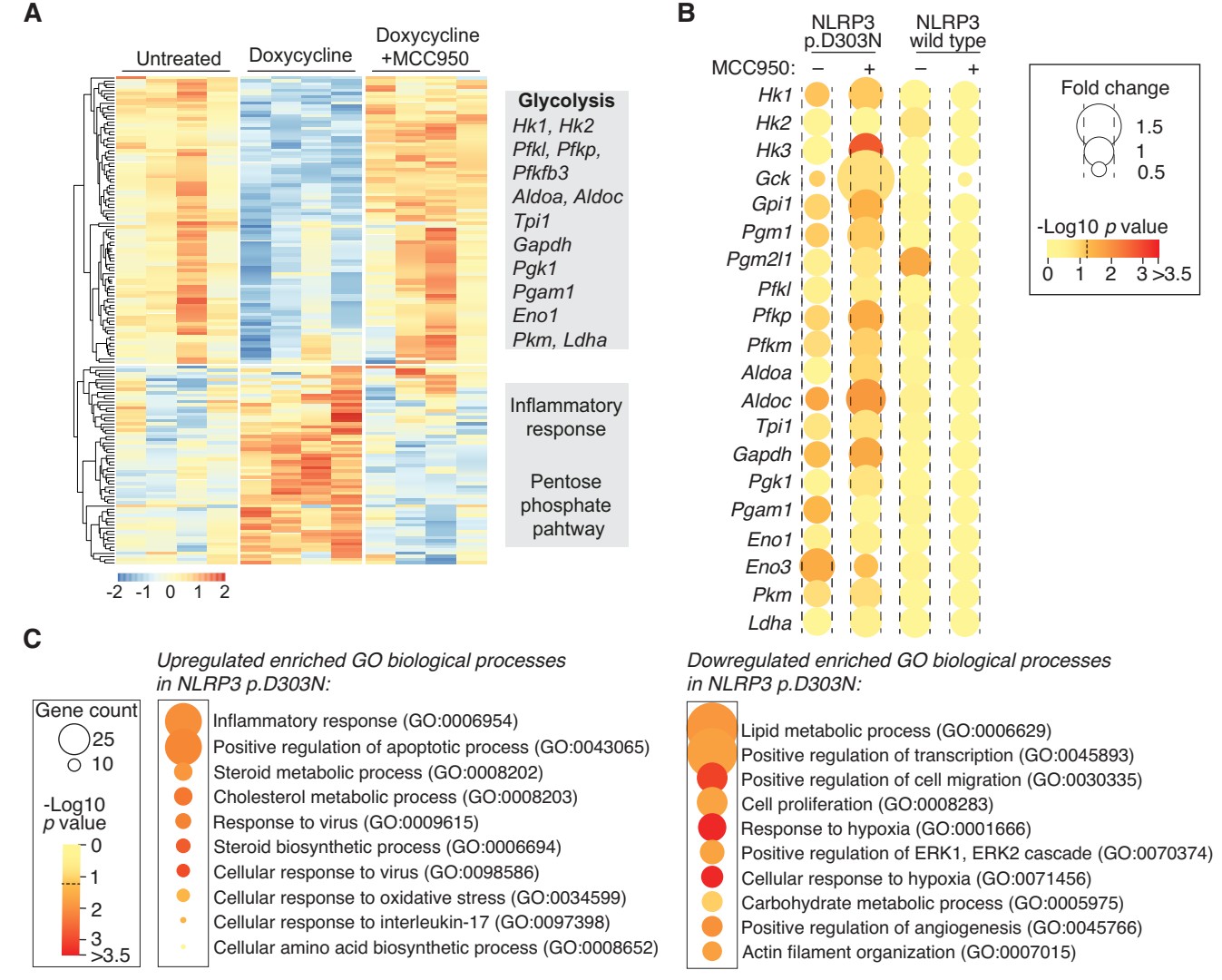

**Fig. 8 | Transcriptomic analysis of macrophages expressing the pathogenic NLRP3 p.D303N variant. A** Gene expression profile of $Nlrp3^{-/-}$ immortalized macrophages (iMos) treated for 16 h with or without doxycycline (1 μg/ml) to induce the expression of the human NLRP3 p.D303N variant, in the absence or presence of MCC950 (10 μM). The gray boxes on the right highlight some of the genes or pathways identified in each trend. The data represented in the graphics are derived from four independent experiments. **B** Relative expression of glycolysis genes from iMos treated as described in **A**, but expressing either wild-type NLRP3 or the p.D303N variant. The data are represented as circles, where the size indicates the fold change of doxycycline treated cells *vs* untreated cells, and doxycycline with

MCC950 treatment *vs* doxycycline, and the color represents the *t*-test two-sided −Log10 *p*-value (the dotted line in the color scale represents *p* = 0.05). The graphics represent data from three to four independent experiments. **C** Gene ontology for biological process enrichment analysis of differentially expressed genes from $Nlrp3^{-/-}$ iMos treated as described in **A**. The data are represented as circles, where the size indicates the gene count for that particular process, and the color represents the −Log10 *p*-value calculated with one-sided Fisher's Exact test with Benjamini−Hochberg correction (the dotted line in the color scale represents *p* = 0.05). The graphics represent data from four independent experiments. Source data are provided as a Source Data file.

induction of IL-1β release, a phenomenon scarcely observed upon constitutive activation of NLRP3 in unprimed cells, despite increased *IL1B* gene expression. This can be attributed to the altered immunometabolism observed during basal NLRP3 activation linked with CAPS-associated variants, which, among others, impairs glycolysis curbing IL-1β production and limiting pathogenic inflammatory flares, while preserving a basal inflammatory status as a defence mechanism against pathogens.

Most of the previously published research on the functional characterization of mutant NLRP3 variants has been conducted by exposing monocytes or macrophages to LPS for a span of 2 to 5 h, followed by measuring the release of IL-1β [i.e., refs. 6,12]. Under these experimental conditions, the release of IL-1β is exacerbated by LPS in samples from CAPS patients compared to those carrying wild type NLRP3, while it is usually undetectable in unprimed cells. A detailed

analysis of additional aspects of NLRP3 inflammasome activation reveal that in basal unprimed cells there is no observation of caspase-1 activation, IL-18 release, processing of GSDMD or pyroptosis[10–16]. These considerations differ from the results of our study. The discrepancies may be due to the brief culture period of the unprimed cells in previous studies, which likely causes only a minimal activation of the NLRP3 inflammasome making it challenging to detect. Supporting this concept, our findings indicate that a low *NLRP3* expression, induced by a 2 h doxycycline incubation, was insufficient to trigger detectable GSDMD cleavage or IL-18 release by using our methods. Given that *NLRP3* and *IL1B* gene expression in myeloid cells is strongly induced following NF-κB activation[18,32], LPS priming is expected to enhance *NLRP3* expression in cells from CAPS patients, yet mutant NLRP3 can constitutively assemble active inflammasomes. Although NLRP3 expression in CAPS patients with active diseases is comparable to that

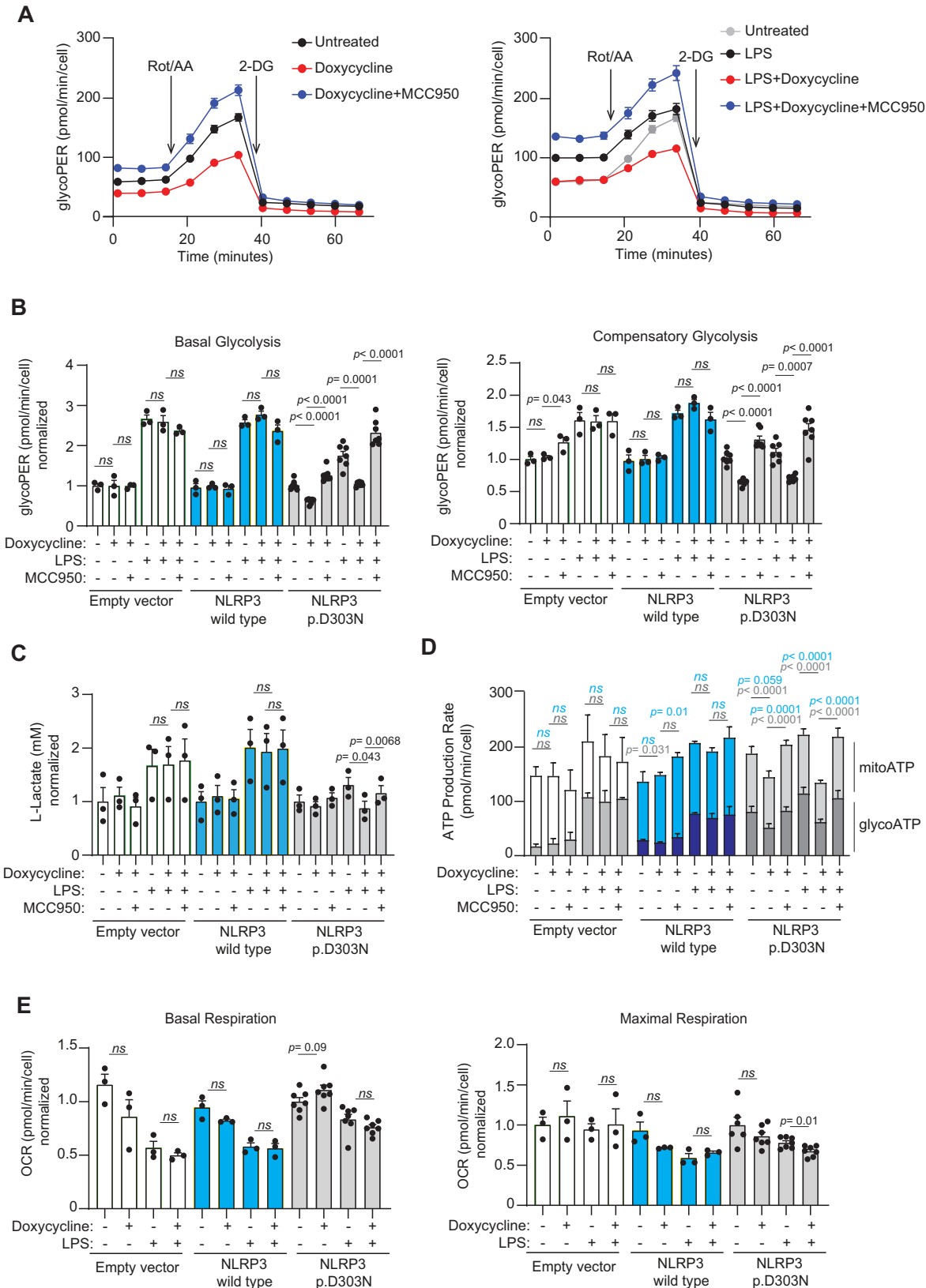

of healthy donors, as per GEO datasets (GSE57253)[30], LPS has been commonly used in previous studies to stimulate IL-1β expression and release from CAPS samples[10–16]. Our recombinant system, which uses NLRP3-deficient immortalized macrophages, effectively mirrors the monocyte behavior seen in CAPS patients and their IL-1β release upon LPS exposure. By adjusting doxycycline levels or incubation durations

we controlled mutant NLRP3 gene expression and we discovered that, even without priming signals, there was a constitutive functional activation of the inflammasome reliant on *NLRP3* gene expression. Consequently, the measurement of IL-1β release in CAPS-related cells should not be the only marker of inflammasome activation in the absence of NF-κB induction. Our study demonstrates that

**Fig. 9 | Decreased glycolytic capacity of macrophages expressing the pathogenic NLRP3 p.D303N variant. A, B** Seahorse analysis of glycolysis in *Nlrp3*[−/−] immortalized macrophages (iMos) treated for 16 h with or without doxycycline (1 μg/ml) to induce the expression of the human NLRP3 p.D303N variant, in the absence or presence of MCC950 (10 μM) and LPS (100 ng/ml). In **B** iMos were not expressing NLRP3 (white) or expressed either the human NLRP3 wild type (blue) or the p.D303N variant (gray). **C** Measurement of L-Lactate in the supernatants from iMos treated as described in **A. D** Seahorse analysis of glycolytic ATP and mitochondrial ATP production rates from iMos treated as described in **A.** The darker columns represent glycolytic ATP, and the lighter columns represent mitochondrial ATP. **E** Seahorse analysis of mitochondria activity measuring basal and

maximal respiration from iMos treated as described in **A.** Data are represented as mean ± SEM; For **A** data are derived from *n* = 7 biological replicates, which are representative of *n* = 4 independent experiments. For **B, D, E** *n* = 3 biological replicates for empty vector and wild type NLRP3 (representative of *n* = 3 independent experiments) and *n* = 7 biological replicates for NLRP3 p.D303N, except *n* = 6 in untreated conditions for maximal respiration in **E** (representative of *n* = 4 independent experiments). For **C** *n* = 3 independent experiments. *t*-test two-sided was used in **C–E** and ordinary one-way ANOVA test was used for **B**; *ns* indicates no significant difference (*p* > 0.05). In **D**, blue represent statistics from mitochondrial ATP and gray represents statistics from glycolytic ATP. Source data are provided as a Source Data file.

macrophages carrying pathogenic NLRP3 variants trigger IL-1β release when treated with various NF-κB activators, including TLR2 agonist Pam3-CSK$_4$. Most importantly, several host-related compounds like palmitate, S100A9 or IL-6 also elicit this response. Interestingly, responses varied among the different NLRP3 mutations we studied; the p.T348M mutation, in particular, resulted in reduced IL-1β release. These findings broaden the range of molecules known to trigger a robust IL-1β–driven response in both active CAPS patients beyond just LPS. Notably, S100A9 and IL-6 levels have been previously reported to be elevated in both CAPS patients and animal models of the disease[30,33–36]. However, the lack of the IL-6 receptor in the murine model does not alter the pathology[37], thus suggesting that while recombinant IL-6 could induce IL-1β release in our immortalized macrophage system, it may not be directly linked to CAPS pathophysiology[37]. This is further supported by the inability of IL-6 to prompt IL-1β release in blood samples from CAPS patients.

The conclusions of our study are reinforced by previous research that identified activation of the NLRP3 in unprimed primary PBMCs from CAPS patients and in recombinant THP-1 cells expressing mutant NLRP3[38–41]. Other studies have also highlighted increased reactive oxygen species production, ATP release and reduced cell survival in unstimulated monocytes from CAPS patients[42–44]. Our research further emphasizes the inherent activity of the NLRP3 inflammasome associated with CAPS. We also found that pathogenic NLRP3 variants exhibiting basal inflammasome activation are modulated by ubiquitination, a known inhibitor of the canonical NLRP3 inflammasome activation[17]. This is consistent with recent insights indicating that pharmacological targeting of NLRP3 deubiquitination inhibits the activation of multiple pathogenic NLRP3 variants[45]. Moreover, other post-transcriptional NLRP3 modifications, like phosphorylation, have been proposed to modulate the gain-of-function behavior of pathogenic NLRP3 variants[46]. Thus, managing post-transcriptional NLRP3 priming at various levels, such as blocking NF-κB activation to downregulate the *NLRP3* gene expression, might offer avenues to treat or prevent inflammatory outbreaks in CAPS patients[40].

While our research uncovered various triggers, including endogenous ones, that modulate the activity of the mutant NLRP3 inflammasome, CAPS are classified as autoinflammatory syndromes[47]. Our data supports this classification, and suggest that it could be due to the expression of a constitutively active NLRP3 inflammasome. Consistently, the systemic expression of NLRP3 with gain-of-function pathogenic variants in a murine CAPS model replicates a sterile inflammation phenotype even without any trigger. These clinical manifestations range from animal growth impediments, sterile skin abscesses, hair loss, arthropathy and osteoporosis, to splenomegaly, lymphopenia, liver fibrosis, saccular-stage lung morphogenesis and peritonitis, all leading to premature death[13,15,35–37,44,48–50]. The inflammatory disease in these animals is driven by various mechanisms, including the ASC/caspase-1 inflammasome, cytokines IL-1β, IL-18, TNF-α and GSDMD-mediated pyroptosis[35,48,50]. Nonetheless, we identified additional secretome regulated by pathogenic NLRP3 variants that could also contribute to the perpetuation of inflammatory flares in CAPS patients. This includes the release of IL-1α, HMGB1 or the

purinergic P2X7 receptor. Interestingly, the plasma concentration of P2X7 receptor has been found to increase in patients during different inflammatory pathologies and correlates positively with the plasma concentration of C-reactive protein[51–54]. Our results suggest that the plasma concentration of P2X7 receptor might also serve as a potential biomarker for CAPS-related inflammation, akin to C-reactive protein[55].

Additionally, the constitutive activation of NLRP3 inflammasomes carrying pathogenic variants dictates a gene expression program that increases inflammatory genes. This includes the upregulation of nuclear receptors 4A1 and 4A2, which although not previously associated with CAPS, control the inflammatory activation of macrophages and have been implicated in rheumatic diseases[56]. CAPS-associated NLRP3 variants also control other genes related to innate immunity (such as lipocalin 2, C-C chemokine receptor type 7, or the zinc finger endoribonuclease Zc3h12a) and IL-17 responses. This is significant since cutaneous manifestations of CAPS are driven by IL-17[13]. Furthermore, the expression of pathogenic NLRP3 variants was associated with upregulated defence mechanisms against viruses, which may explain the clinical observation that untreated CAPS patients are not prone to suffer from infections. Conversely, cellular homeostatic processes were downregulated upon expression of CAPS-associated NLRP3 variants.

Advances in the field of immunometabolism have revealed how metabolic processes are crucial to regulate the inflammatory function of macrophages, with impact on the progress of diseases[28,29]. Stimulation of macrophages with LPS triggers metabolic reprogramming, leading to enhanced glycolysis and an upregulated pentose phosphate pathway, while reducing the tricarboxylic acid cycle[28]. Our study determined that the basal activation of the CAPS-associated NLRP3 inflammasome significantly alters cellular metabolites, including lipids, lipids-like molecules, amino acids and their intermediates. These findings are further supported by the upregulation of biological processes and genes associated with these pathways. Interestingly, echoing the metabolic transition from oxidative metabolism to glycolysis seen post-LPS stimulation in macrophages[28,29], we found that certain key glycolytic genes (such as *PFKFB3* or *BPGM*) were upregulated in blood cells from CAPS patients collected during active disease, while others (*ENO3*) were downregulated. On a cellular level, the basal activation of the CAPS–associated NLRP3 inflammasome resulted in decreased expression of glycolytic genes and a reduction in glycolysis flux and glycolytic ATP production. This led to a specific limitation of IL-1β production, potentially impairing IL-1–driven inflammatory flares, while maintaining IL-18 release and a basal innate immune activation in CAPS, which is crucial for protection against pathogens. Beyond the downregulation of glycolytic genes, it is known that caspase-1 activation can also reduce glycolysis by processing key glycolytic enzymes[57].

One limitation of this study is the potential discrepancy between the gene expression and metabolic programs between immortalized mouse macrophages expressing the p.D303N NLRP3 variant and those from primary mouse macrophages or human monocytes/macrophages, both in vitro and in vivo. Furthermore, our study solely focused on the metabolism affected by one pathogenic NLRP3 variant

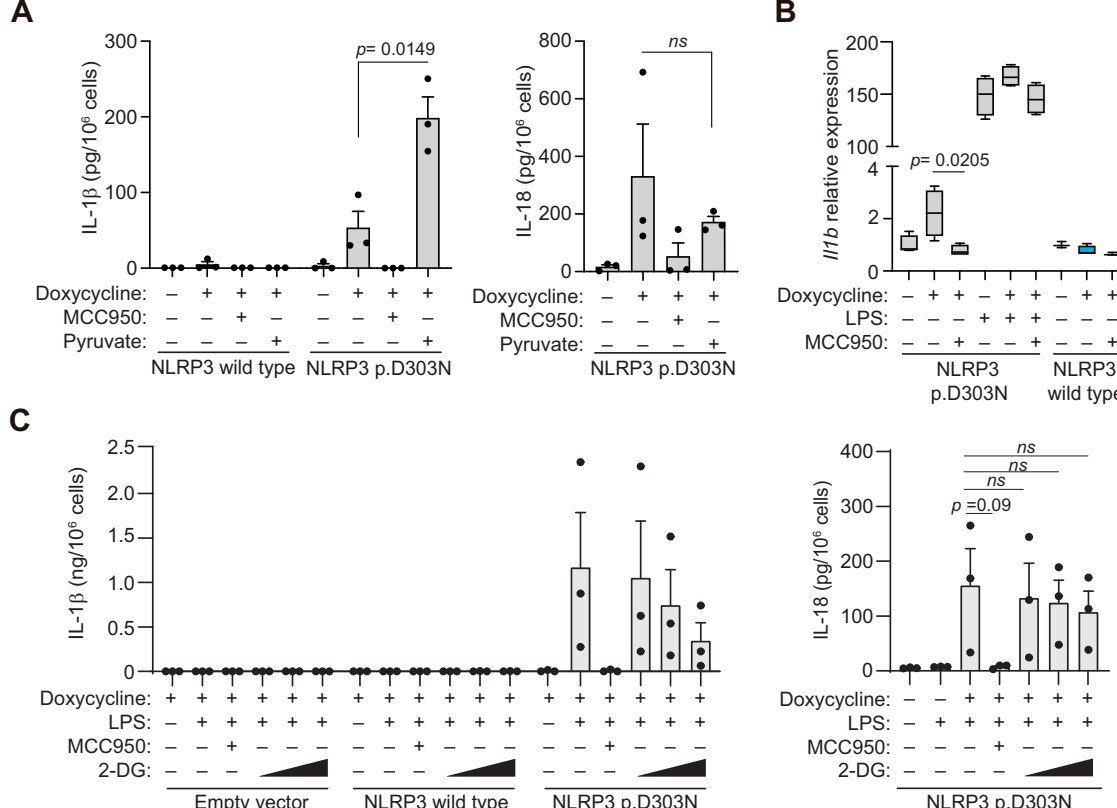

**Fig. 10 | Glycolysis modulates the release of IL-1β, but not IL-18, in macrophages expressing NLRP3 p.D303N. A** ELISA for IL-1β and IL-18 from *Nlrp3*$^{-/-}$ immortalized macrophages (iMos) treated for 16 h with or without doxycycline (1 μg/ml) to induce the expression of either the human wild type NLRP3 or the p.D303N variant, in the absence or presence of MCC950 (10 μM) and pyruvate (10 mM). The data are derived from *n* = 3 independent experiments. **B** Box plot of *Il1b* mRNA expression from *Nlrp3*$^{-/-}$ iMos treated as described in **A**, but were expressing either wild type NLRP3 or the p.D303N variant and were treated with or without LPS (100 ng/ml). Middle line depicts the mean, error bars represent minimum and maximum, and bounds of box represent the 25th to 75th percentile respectively; graphic represents data from *n* = 4 independent experiments, except for NLRP3 wild type untreated and doxycycline+MCC950 treatment that were *n* = 3 independent experiments. **C** ELISA for IL-1β and IL-18 from *Nlrp3*$^{-/-}$ iMos expressing either the human wild type NLRP3 or the p.D303N variant induced after treatment for 4 h with doxycycline (1 μg/ml) in the absence or presence of LPS (100 ng/ml), MCC950 (10 μM) and 2-DG (0.1, 0.5 and 1 mM). The data are derived from *n* = 3 independent experiments. Data are represented as mean ± SEM; a *t*-test two-sided was used in **A**–**C**; *ns* indicates no significant difference (*p* > 0.05). Source data are provided as a Source Data file.

(p.D303N) expressed in immortalized macrophages. Despite this, we observed that the metabolic profile and gene expression in monocytes from human CAPS patients carrying additional pathogenic NLRP3 variants were similar to those in immortalized macrophages expressing the p.D303N human NLRP3 variant. This observation suggests that a reduction in glycolysis could limit IL-1β–derived inflammatory flares in CAPS.

Overall, our study reveals that gain-of-function pathogenic NLRP3 variants, which are associated with CAPS, lead to the formation of a constitutively active inflammasome dependent on NLRP3 expression. This activation occurs even in the absence of triggers or cell priming, and it influences the immunometabolism of CAPS patients, specifically by hindering glycolysis and consequently limiting IL-1β production and inflammatory flares, but orchestrating a specific inflammatory program against pathogens. Active pathogenic inflammasomes can be further modulated by deubiquitination and signals associated with pathogens or sterile triggers that activate the NF-κB pathway, intensifying inflammasome-related responses, including IL-1β production and CAPS flares.

## Methods

This research complies with all relevant ethical regulations, and the study protocol was approved by the ethical committee of the University Clinical Hospital Virgen de la Arrixaca (Murcia, Spain).

### Reagents and buffers

Different reagents and their sources used in this study were: ATP, nigericin, BSA fatty acid-free and low endotoxin, palmitic acid, doxycycline, MCC950 (CP-456773), and the authentic standards citric acid, malic acid, fumaric acid, succinic acid, ketoglutaric acid, pyruvic acid and lactic acid were all purchased from Sigma-Aldrich; Palmitic acid was conjugated with BSA as described previously[58]; ultrapure *E. coli* LPS serotype 0111:B4 (tlrl-3pelps) and Pam3CSK$_4$ (tlrl-pms) from InvivoGen; recombinant mouse IL-1Ra protein (HY-P72566) from MedChemExpress; recombinant mouse IL-6 protein (RMIL6I) from Invitrogen; recombinant mouse S100A9 protein (2065-s9), recombinant human S100A9 protein (9254-s9), recombinant human IL-6 protein (7270-IL/CF) from R&D systems; Pyruvate (L064-100) from Biowest; 2-deoxy-D-glucose (2-DG) (14325), ubiquitin isopeptidase inhibitor I G5 (21006), PR-619 (16276) and b-AP15 (11324) from Cayman Chemical Company. Acetonitrile and water 0.1% (v/v) formic acid were from J.T. Baker, and formic acid was obtained from Panreac. Ultrapure water filter through Milli-Q system (Millipore Corp).

### Human samples

Samples and data from humans volunteers included in this study, who gave written informed consent, were stored in the Biobanco en Red de la Región de Murcia, BIOBANC-MUR, registered on the National Registry of Biobanks with registration number B.0000859, and were used

following standard operating procedures with appropriate approval of the Ethical Committee of the Clinical University Hospital Virgen de la Arrixaca (Murcia, Spain). Gender was not considered in this study, as CAPS is a rare condition and scare patients are available to equilibrate among genders (Supplementary Table 1). Whole blood samples were collected in EDTA anticoagulated tubes from healthy donors ($n = 9$) and from individuals with CAPS (MWS) carrying the NLRP3 p.R260W ($n = 1$), p.D303N ($n = 1$), p.T348M ($n = 1$) and p.A439T variant ($n = 4$). Samples were all used to obtain the results of this study. CAPS patients were with inactive disease under IL-1 blocking therapy by the time blood was extracted (Supplementary Table 1). Human peripheral blood mononuclear cells (PBMCs) were isolated using Ficoll Histopaque-1077 (Sigma-Aldrich) and cultured in Opti-MEM Reduced Serum Media (Gibco) at 37 °C and 5% $CO_2$. Cells were stimulated with LPS (100 ng/ml), Pam3-CSK$_4$ (1 μg/ml), recombinant human S100A9 protein (500 ng/ml), recombinant human IL-6 protein (500 ng/ml), palmitate-BSA conjugated (1 mM) and MCC950 (10 μM) during 2, 4 and 6 h in Opti-MEM. In some experiments, cells were treated with G5 and b-AP15 (both at 5 μM) in absence or presence of LPS for 6 h.

### Monocyte purification
Monocytes were isolated from PBMCs using the MagniSort™ Human CD14 Positive Selection Kit (Thermofisher) and $2 \times 10^6$ monocytes were cultured in Opti-MEM Reduced Serum Media (Life Technologies) at 37 °C and 5% $CO_2$ in presence or absence of 500 ng/ml LPS for 2 h. Control cultures without cells were run in parallel as metabolic control.

### Immortalized mouse macrophages generation and stimulation
For doxycycline-inducible expression of NLRP3 variants in $Nlrp3^{-/-}$ immortalized mouse macrophages (a gift from I. Hafner-Bratkovič, National Institute of Chemistry, Ljubljana, Slovenia), we used the Tet-ON retroviral system (Clontech) with NLRP3-YFP wild type or p.D303N, p.T348M and p.R260W mutants[22,25]. Cells were treated with doxycycline (0.25, 0.5 and 1 μg/ml) to express NLRP3, in the presence or absence of MCC950, ATP, MSU, LPS, Pam3-CSK$_4$, S100A9, IL-6, or palmitate at different concentrations and incubation times as indicated in the figure legends. In some experiments, the cells were also incubated with pyruvate (10 mM), the following deubiquitinase inhibitors G5 (5 μM), PR-619 (10 μM) and b-AP15 (5 μM), or with the glycolysis inhibitor 2-DG (0.1, 0.5 or 1 mM).

### Ultra-performance liquid-chromatography-electrospray ionization-quadrupole time-of-flight mass-spectrometry (UPLC-ESI-QTOF-MS) untargeted metabolomics analysis
After stimulation, cells were washed with PBS twice and resuspended in 2 ml (monocytes) or 800 μl (macrophages) of MeOH (PanReac AppliChem) to obtain polar metabolites. Four replicates of each group were processed for human monocytes and three replicates for immortalized macrophages. Samples were homogenised in a vortex for 1 min, and then centrifuged for 10 min at $6000 \times g$ at 4 °C (Thermo Scientific). The resultant supernatants were filtered through a 0.22 μm PVDF filter prior the injection in the UPLC-ESI-QTOF-MS system. Four replicates each one from an individual donor of each group were processed in monocytes. The analyses were performed by an Agilent 1290 Infinity LC system coupled to a 6550 Accurate-Mass Quadrupole time-of-flight (QTOF) (Agilent Technologies) using an electrospray interface (Jet Stream Technology). Chromatographic separation was carried out on a reversed-phase C18 column (Poroshell 120, $3 \times 100$ mm, 2.7 μm pore size) at 30 °C, using water with 0.1% formic acid (Phase A) and acetonitrile with 0.1% formic acid (Phase B) as mobile phases with a flow rate of 0.4 ml/min. The following gradient was used: 0–10 min, 1–18% phase-B; 10–16 min, 18–38% phase-B; 16–22 min, 38–95% phase-B. Finally, the phase B content was returned to the initial conditions (1%) for 1 min and the column re-equilibrated then for 5 min. The mass analyser was operated in negative mode

under the following conditions: gas temperature 180 °C, drying gas 12 l/min, nebulizer pressure 45 psig, sheath gas temperature 350 °C, sheath gas flow 11 l/min, capillary voltage 3500 V, nozzle voltage 250 V, fragmentor voltage 350 V, and octapole radiofrequency voltage 250 V. Data were acquired over the $m/z$ range of 50–1700 at the rate of 2 spectra/s. The data were acquired in negative and positive polarities. The raw data files were acquired by the UPLC-ESI-QTOF-MS system in profile file mode and were exported to MZmine software (Version 2.53, © 2005-2015 MZmine Development Team) to create the data matrix. The raw data generated of both polarities were pre-processed separately by a batch set of parameters including the mass detection, chromatogram builder and deconvolution and alignment algorithm. The data matrices were then merged by MZmine and exported to Mass Profiler professional (MPP, Agilent technologies) and Metaboanalyst 5.0 online platform (www.metaboanalyst.ca) for parallel data management. Data matrices were processed including log transformation and auto scaling prior to univariate and multivariate analysis[59]. The multivariate analysis PCA (Principal component analysis) was performed to study the total data variation of the data samples groups and evaluate the group trends. The calculated PCA model was built based on 16 samples and three components in monocytes, or 12 samples and three components in macrophages. The first two principal components PC1 and PC2 explained 25.2% and 9.6% of the total variability for monocytes, and 29.5% and 15.9% of the total variability in case of macrophages. Both PCA models showed the differences according to the total covariance between the study groups. The univariate analysis was performed by MPP software after the multivariate analysis evaluation. Data treatment through MPP software included filters by frequency of the data matrix to reduce the sample variability within each study group. ANOVA and $t$-test unpaired (corrected $p$-value cut-off: 0.05; $p$-value computation: asymptotic; multiple testing correction: Benjamini-Hochberg) statistics analysis was applied to the data matrix to filter significant entities along the different sample groups. The final list of features was used for metabolite identification with databases according to the exact mass and therefore achieving level 2 of identification[60]. Metabolomic data obtained in this study has been deposited in MetaboLights under the accession code MTBLS7872.

### UPLC-ESI-QTOF-MS targeted metabolomics analysis of glycolysis and TCA cycle
The analyses were set up according to Körver–Keularts et al.[61] with modifications. The UPLC-ESI-QTOF-MS system used for the targeted approach was the same as for untargeted except the reversed-phase column. Chromatographic separation was carried out on a Acquity C18 column UPLC HSS T3 1.8 μm $2.1 \times 100$ mm at 22 °C, using water with 0.1% formic acid (Phase A) and 95% acetonitrile/1% water with 0.1% formic acid (Phase B) as mobile phases with a flow rate of 0.45 ml/min. The following gradient was used: 0–2 min, 0% phase-B; 2–4 min, 0–15% phase-B; 4–10 min, 15–45% phase-B; 10–13 min, 45–100% phase-B; 13–20 min, 100% phase-B; 20–25 min, 0% phase-B; 25–35 min, phase-B. The mass analyser was operated in negative mode under the following conditions: gas temperature 150 °C, drying gas 15 l/min, nebulizer pressure 35 psig, sheath gas temperature 400 °C, sheath gas flow 12 l/min, capillary voltage 3500 V, nozzle voltage 500 V, fragmentor voltage 100 V, and octapole radiofrequency voltage 750 V. Data were acquired over the $m/z$ range of 50–1200 at the rate of 3.5 spectra/s. The data were acquired in negative mode. The raw data files were acquired by the UPLC-ESI-QTOF-MS system in profile file mode and were exported to MassHunter Qualitative Analysis software (Version 10.0, Agilent Technologies) for metabolite identification.

### Transcriptome and gene expression analysis
Data from RNA sequencing from total blood cells from healthy pediatric donors ($n = 5$), pediatrics CAPS patients carrying the NLRP3 p.G569R variant with active disease (NOMID/CINCA) prior to anakinra

treatment ($n = 7$), and the same CAPS patients with inactive disease following anakinra treatment (GSE57253)[30] were analyzed with the package DESeq2 (R studio, Posit, PBC) to calculate differential inflammatory and metabolic gene expression. Data from healthy human PBMCs ($n = 26$-$30$) in vitro treated or not during 4 h with LPS (10 ng/ml) (GSE42606)[62] was analyzed with the GEO2R tool with the package Limma (R studio, Posit, PBC) to calculate differential inflammatory and metabolic gene expression. Data from total blood cells from healthy donors ($n = 6$) and CAPS patients with active disease ($n = 3$) (GSE17732)[31] was analyzed with the GEO2R tool with the package Limma (R studio, Posit, PBC) to calculate differential metabolic gene expression.

Total RNA was extracted from $Nlrp3^{-/-}$ immortalized mouse macrophages expressing NLRP3 wild type or NLRP3 p.D303N using the RNeasy kit (74104, Qiagen) following manufacturer instructions. Upon extraction, total RNA was quantified with Qubit RNA BR Assay (Invitrogen), and RNA integrity was analysed with TapeStation 4200 (Agilent Technologies). All samples showed RIN > 8. cDNA synthesis and library generation were performed using Stranded mRNA Prep Kit (Illumina) following the manufacturer's instructions. RNA-seq libraries were sequenced using Paired-End 200 bases (PE200) sequencing chemistry on NextSeq 2000 Sequencing Systems (Illumina). Sequencing results were analysed using package DESeq2 (R studio, Posit, PBC) for differential analysis of gene expression between two groups. Data generated in this study has been deposited in GEO under accession code GSE246713. In this case, the Wald test was used, and data were corrected with the Benjamini-Hochberg procedure to obtain an adjusted $p$-value. For group representation, all data were normalized using vst DESeq2 function. Enrichment analysis of gene ontology (GO) was determined using the DAVID tool v2021[63,64], the Benjamini−Hochberg method are used to correct the $p$-values, gene sets with adjusted $p < 0.05$ were significantly enriched.

### HEK293T cell culture and transfection

HEK293T cells (CRL-11268, American Type Culture Collection) were maintained in Dulbecco's modified Eagle's medium (DMEM)/ F-12 (1:1) (Lonza) supplemented with 10% foetal calf serum (FCS) (Life Technologies), 2 mM GlutaMAX (Life Technologies), and 1% penicillin-streptomycin (Life Technologies). Cells were maintained at 37 °C in a humidified 5% $CO_2$ incubator. Lipofectamine 2000 (Life Technologies) was used for the transfection of $7 \times 10^5$ HEK293T according to the manufacturer's instructions using 0.1 µg of ASC-RFP plasmid and 0.1 µg of YFP-NLRP3-Luc plasmid and/or YFP-NLRP3 D303N-Luc plasmid[22,55].

### Flow cytometry

Intracellular ASC-speck formation in human monocytes was evaluated by seeding 50 µl of individuals' whole blood samples in polystyrene flow cytometry tubes (Falcon) with RPMI 1640 medium (Lonza) containing 10% FCS and 2 mM Glutamax. Following treatments with inhibitor or triggers, cells were stained for the detection of ASC specks by Time-of-Flight Inflammasome Evaluation (TOFIE)[65,66] using the PE conjugated mouse monoclonal anti-ASC antibody (clone HASC-71, catalog 653903, Biolegend, 1:500). Monocytes were gated using the FITC conjugated mouse monoclonal anti-CD14 antibody (clone M5E2, catalog 557153, BD Biosciences, 1:10) and using the PE-Cy7 conjugated mouse monoclonal anti-CD16 antibody (clone 3G8, catalog 557744, BD Biosciences 1:10). Intracellular ASC-RFP-speck formation in HEK293T cells was evaluated after 24 h post-transfection by TOFIE in different gates with increasing mean fluorescence intensity for NLRP3-YFP (Supplementary Fig. 2D). Human blood samples were analysed by flow cytometry using FACS Canto (BD Biosciences) and for HEK293T cells using LSRFortessa (BD Biosciences), for both cases the FCS express software (De Novo Software) was used.

### Fluorescence microscopy

Poly-L-lysine coated coverslips (Corning) were used to seed cells immortalized mouse macrophages $Nlrp3^{-/-}$ expressing the different variants of NLRP3. Cells were washed after stimulation once with sterile PBS buffer (Gibco) and fixed for 15 min at room temperature with 4% paraformaldehyde (Electron Microscopy Science) and then were washed four times with PBS. Cells were blocked with 0.5% bovine serum albumin (Sigma-Aldrich) and permeabilized with 0.1% Triton X-100 (Sigma-Aldrich) for 40 min at room temperature. Then, cells were incubated for 16 h at 4 °C with the primary rabbit polyclonal antibody anti-ASC (N-15)-R (catalog sc-22514-R, Santa Cruz, 1:500). Cells were washed tree times for 10 min each and then incubated for 1 h with donkey anti-rabbit alexa-647 antibody (catalog A31573, Life Technologies, 1:800). Then cells were washed three times for 10 min each and nuclei were stained with DAPI (1:10,000) for 10 min and coverslips were mounted on slides with mounting medium (Dako). Images were acquired with a Nikon Eclipse $Ti$ microscope equipped with a 20× S Plan Fluor objective (numerical aperture, 0.45) and a digital Sight DS-QiMc camera (Nikon) and 387 nm/447 nm, 543 nm/593 nm filter sets (Semrock), and the NIS-Elements AR software (Nikon). Images were analysed with ImageJ (US National Institutes of Health). Quantification of ASC specking macrophages was done in 4 fields of view per condition and at least from $n = 3$ different experiments, counting at least a total of 1100 cells per condition.

### Western blot

Immortalized mouse macrophages and HEK293T were lysed after stimulation in ice-cold lysis buffer (50 mM Tris-HCl pH 8.0, 150 mM NaCl, 2% Triton X-100) supplemented with 100 µl/ml of protease inhibitor mixture (Sigma-Aldrich) for 30 min on ice and were then centrifugated at 13,000 × $g$ for 10 min at 4 °C. Cells lysates were quantified by Bradford assay (Sigma-Aldrich) and results were read in a Synergy Mx (BioTek) plate reader at 595 nm. Cell supernatants were collected, centrifuged at 600 × $g$ for 5 min at 4 °C and the cell-free supernatants were concentrated using a 10 kDa cut-off column (UFC501024, Merk-Millipore). Cells lysates and concentrated supernatants were denaturalized with Laemmli buffer (Sigma-Aldrich) and heated at 95 °C for 5 min. 40 µg of cell lysates or from 0.5 to 1 ml of concentrated supernatants were resolved in a 12% precast Criterion polyacrylamide gels (Biorad) and transferred to nitrocellulose membranes (Biorad) by electroblotting. Membranes were hybridized with the following primary antibodies: anti-NLRP3 mouse monoclonal (Cryo-2 clone, catalog AG-20B-0014, Adipogen, 1:1000), anti-Caspase 1 (p20) mouse monoclonal (Casper-1, catalog AG-20B-0042, Adipogen, 1:1000), anti-GSDMD rabbit monoclonal (EPR19828, catalog ab209845, Abcam, 1:2500), anti-IL-1β rabbit polyclonal (H-153; catalog sc-7884, 1:1000) and horseradish peroxidase (HRP)-anti-β-actin (C4; catalog sc-47778HRP, Santa Cruz, 1:10,000) and appropriate secondary antibodies: horseradish peroxidase anti-IgG rabbit (catalog NA9340V, Cytiva, 1:5000) and horseradish peroxidase anti-IgG mouse (catalog NA9341V, Cytiva, 1:5000). Membranes were revealed using ECL Plus reagents (Cytiva) in a ChemiDoc Imaging System (BioRad).

### Seahorse assay

$Nlrp3^{-/-}$ immortalized macrophages expressing or not NLRP3 p.D303N variant and NLRP3 wild type were seeded at 25.000 cells/well in Seahorse adherent 96-well plate (Agilent Technologies) in DMEM high glucose medium (Biowest) without FBS. Cells were treated with doxycycline (1 µg/ml), LPS (100 ng/ml), MCC950 (10 µM), IL-1Ra (100 ng/ml) or 2-DG (0.1, 0.5, 1 mM) for 4 h or 16 h (as indicated in the figure legend) at 37 °C and 5% $CO_2$. After stimulation, cell media was discarded, and the cells were incubated with 180 µl DMEM seahorse medium supplemented with 5 mM glucose, 1 mM pyruvate and 2 mM glutamine (Agilent Technologies). The plate was incubated for 45 min at 37 °C without $CO_2$. Glycolytic rate assay, ATP real-time rate

assay, and Cell Mito-stress kits (Agilent Technologies) were used for experiments according to the manufacturer's instructions using an XF96e Analyzer (Agilent Technologies). After reading, the nuclei of the cells were stained with 3 mM Hoechst solution (Sigma-Aldrich) to normalize the number of cells in the calculated OCR and ECAR values. Results were collected with the Wave software version 2.6 (Agilent Technologies).

## L-Lactate determination

To measure glycolysis, the presence of L-lactate in cell-free supernatants was measured using the Glycolysis assay kit (BA0086, Assay Genie) according to the manufacturer's instructions, the samples were read in a Synergy Mx plate reader (BioTek) at 565 nm.

## Lactate dehydrogenase assay

To measure cell death, the LDH present in cell-free supernatants was detected using the Cytotoxicity Detection kit (Roche) according to manufacturer instructions, the reaction was read in a Synergy Mx (BioTek) plate reader at 492 nm and corrected at 620 nm.

## NF-κB reporter assay

To evaluate NF-κB activation we used the RAW264.7 macrophage cell line expressing a secreted embryonic alkaline phosphatase (SEAP) reporter gen under the control of the NF-κB promoter (Invivogen), which were cultured according to manufacturer instructions. Cells were treated with ATP (5 mM), MSU (200 μg/ml), LPS (100 ng/ml), Pam3-CSK$_4$ (1 μg/ml), S100A9 (500 ng/ml), IL-6 (500 ng/ml), or palmitate (1 mM) for 4 h, in some experiments the LPS treatment was incubated for 6 h in the presence or absence of G5 (5 μM), PR-619 (10 μM) and b-AP15 (5 μM). Cell supernatants were collected, centrifuged at $600 \times g$ for 5 min at 4 °C and the cell-free supernatants were mixed with the Quanti-blue solution (rep-qbs, Invivogen) according to manufacturer instructions. The reaction was read in a Synergy Mx (BioTek) plate reader at 622 nm.

## ELISA

Cell-free supernatants were collected and the following ELISA kits were used according to manufacturer instructions: mouse IL-1β (88-7013-22, Invitrogen), mouse TNF-α (88-7324-22, Invitrogen), mouse IL-18 (BMS618-3, Invitrogen), mouse P2X7 (OKEH05605, Aviva Systems Biology), mouse Cathepsin B (ab119585, Abcam), mouse Annexin A1 (ab264613, Abcam), mouse IL-1α (MLA00, R&D Systems), mouse HMGB1 (ARG81310, Arigo), mouse CD14 (D4982, R&D Systems), mouse Cystatin B (Cstb, orb565550, Biorbyt), mouse CD206 (orb546630, Biorbyt), human IL-18 (7620, MBL), human galectina 3 (BMS279-4, Invitrogen), human IL-1β (BMS224INST, Invitrogen) and human TNF-α (DTA00D, R&D Systems). ELISA were read in a Synergy Mx (BioTek) plate reader at 450 nm and corrected at 540 and 620 nm.

## Quantitative reverse transcriptase-PCR analysis

Total RNA was extracted and purified from immortalized macrophages using the RNeasy kit (74104, Qiagen) following manufacturer instructions and quantified in a NanoDrop 2000 (Thermo Fisher). RNA was reverse transcribed using the iScript™ cDNA Synthesis kit (1708891, BioRad) according to manufacturer instructions. Quantitative PCR was done in an iQTM 5 Real Time PCR System (BioRad) with SYBR Green mix (Takara) and predesigned KiCqStart primers for mouse *Il1b, Tnfa, Il18, Pycard, Gsdmd, Nlrp3, Nek7* and *Casp1* (Sigma-Aldrich). Gene expression was normalized with *Gapdh* as endogenous control.

## Statistics and reproducibility

Statistical analyses were performed using GraphPad Prism 9 (Graph-Pad Software Inc) or R-Studio (Posit, PBC). Normality of the samples was determined with D'Angostino and Pearson omnibus K2 normality test. Outliers from data sets were identified by the ROUT method with

$Q = 1\%$ and were eliminated from the analysis and representation. All data are shown as mean values and errors bars represent standard error (SEM). For two-group comparisons, the Mann-Whitney U test was used in non-parametric data, and the *t*-test was used in parametric data. The comparisons of multiple groups were analysed by Kruskal–Wallis for non-parametric data and ANOVA analyses were used for parametric data. For dataset GSE42606 and GSE17732, gene expression comparison was performed with *t*-test. For dataset GSE57253 and immortalized mouse macrophages RNA sequencing the Wald test was used. In all datasets, data were corrected with the Benjamini-Hochberg procedure to obtain adjusted *p*-value. The *p* values are indicated with asterisks and ranges are noted in the figure legends, for $p > 0.05$ was considered not significant (*ns*). The exact *n* number of independent experiments for the Fig. 4 panels A and B were as follows:

For Fig. 4A, for empty vector transduced macrophages $n = 3$ independent experiments, except $n = 4$ independent experiments for untreated, LPS and palmitate; for wild type NLRP3 transduced macrophages $n = 3$ independent experiments, except $n = 4$ independent experiments for untreated and LPS; for NLRP3 p.D303N transduced macrophages $n = 4$ independent experiments, except $n = 5$ independent experiments for untreated and LPS and $n = 3$ independent experiments for MCC950 treatments; for NLRP3 p.R260W transduced macrophages $n = 4$ independent experiments, except $n = 3$ independent experiments for palmitate and S100A9; for NLRP3 p.T348M transduced macrophages $n = 5$ independent experiments for untreated and LPS, $n = 4$ independent experiments for IL-6 and Pam3CSK$_4$, and $n = 3$ independent experiments for palmitate and S100A9 (each independent experiment is represented by a different symbol).

For Fig. 4B, for empty vector transduced macrophages $n = 3$ independent experiments, except $n = 4$ independent experiments for untreated, LPS and palmitate; for wild type NLRP3 transduced macrophages $n = 3$ independent experiments, except $n = 4$ independent experiments for untreated and LPS; for NLRP3 p.D303N transduced macrophages $n = 4$ independent experiments for palmitate, S100A9 and IL-6, $n = 5$ independent experiments for untreated and LPS and $n = 3$ independent experiments for ATP, Pam3CSK$_4$, MSU and all MCC950 treatments; for NLRP3 p.R260W transduced macrophages $n = 4$ independent experiments, except $n = 3$ independent experiments for palmitate and S100A9; for NLRP3 p.T348M transduced macrophages $n = 5$ independent experiments, except $n = 3$ independent experiments for palmitate and S100A9 (each independent experiment is represented by a different symbol).

## Inclusion and ethics

We support inclusive, diversity, and equilibrate conduct of research. Whenever possible, we worked to ensure gender balance, ethnic and other types of diversity in the recruitment of human subjects. The protocol to include samples and data from humans included in this study was approved by the Ethical Committee of the Clinical University Hospital Virgen de la Arrixaca (Murcia, Spain).

## Reporting summary

Further information on research design is available in the Nature Portfolio Reporting Summary linked to this article.

## Data availability

The metabolomic data generated in this study have been deposited in the MetaboLights database under accession code MTBLS7872. The RNAseq data generated in this study have been deposited in the GEO database under accession code GSE246713. The published RNAseq data used in this study are available in the GEO database under accession codes GSE57253, GSE42606 and GSE17732. Source data are provided with this paper.

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

## Acknowledgements

We thank I. Hafner-Bratkovič (National Institute of Chemistry, Slovenia) for immortalized macrophages with Tet-ON system, M.C. Baños and A.I. Gomez (IMIB, Murcia, Spain) for technical assistance with molecular and cellular biology, Genomics central facility (IMIB, Murcia, Spain) for sequencing support, F. Perez-Sanz (Bioinformatics service, IMIB, Murcia, Spain) for RNA sequencing analysis, C. de Torre-Minguela (IMIB, Murcia, Spain) for initial help with metabolomics experiments design and the members of the Pelegrín's laboratory for comments and suggestions thought the development of this project. We are also particularly grateful for the generous contribution of the patients and the collaboration of Biobank Network of the Region of Murcia, BIOBANC-MUR. BIOBANC-MUR is supported by the "Instituto de Salud Carlos III (PT23/00026), by "Instituto Murciano de Investigación Biosanitaria Virgen de la Arrixaca, IMIB" and by "Consejería de Salud de la Comunidad Autónoma de la Región de Murcia. This work was supported by grants from MCIN/AEI/10.13039/501100011033 (grants PID2020-116709RB-I00, BG20/00060, CNS2022-135101 and RED2022-134511-T to PP), *FEDER/Ministerio de Ciencia, Innovación y Universidades – Agencia Estatal de Investigación* (grant SAF2017-88276-R to PP), Fundación Séneca (grants 20859/PI/18 and 21897/PI/22 to PP), the Instituto Salud Carlos III (grants DTS21/00080 and AC22/00009 to PP), the EU Horizon 2020 project PlasticHeal (grant 965196 to PP), the Spanish Ministry of Science, Innovation and Universities cofunded by the European Regional Development Fund / Agencia Estatal de Investigación (grant RTI2018-096824-B-C21 to JIA) and the Spanish Ministry of Science and Innovation / Agencia Estatal de Investigación (grant PID2021-125106OB-C31 to JIA). L.H.-N. was supported by the fellowship 21214/FPI/19 (Fundación Séneca, Región de Murcia, Spain) and CM-L by was funded by the fellowship PRE2018-086824 (Ministerio economía y competitividad).

## Author contributions

C.M.-L. performed the experimental work; L.H.-N. supported with human sample experimentation and performed flow cytometry experiments; D.A.-B., A.T.-A. supported with the generation of mutant NLRP3 and immortalized macrophage lines; J.R.M., C.V., S.B.-R., J.I.A. obtained CAPS patient samples and clinical history; C.J.G., F.V. and F.A.T.-B. performed and analyzed metabolomic assays; C.M.-L. and P.P. analyzed the data, interpreted results, conceived the experiments, prepared the figures and paper writing; P.P. conceived the project, provided funding and overall supervision of this study.

## Competing interests

P.P. declares that he is an inventor in a patent filled on March 2020 by the *Fundación para la Formación e Investigación Sanitaria de la Región de Murcia* (PCT/EP2020/056729) for a method to identify NLRP3-immunocompromised sepsis patients. P.P., L.H.-N. and D.A.-B. are co-founders and have shares in Viva in vitro diagnostics SL, but declare that the research was conducted in the absence of any commercial or financial relationships that could be construed as a potential conflict of interest. The remaining authors declare no competing interests.

## Additional information

[1]Molecular Inflammation Group, Instituto Murciano de Investigación Biosanitaria Pascual Parrilla–IMIB, Murcia, Spain. [2]Quality, Safety and Bioactivity of Plant-Derived Foods, Centro de Edafología y Biología Aplicada del Segura-Consejo Superior de Investigaciones Científicas (CEBAS-CSIC), Murcia, Spain. [3]Department of Internal Medicine, Hospital Vall Hebron, Barcelona, Spain. [4]Department of Rheumatology, Hospital Virgen de la Macarena, Sevilla, Spain. [5]Department of Immunology, Hospital Clínic, Barcelona, Spain. [6]Institut d'Investigacions Biomèdiques August Pi i Sunyer, Barcelona, Spain. [7]School of Medicine, Universitat de Barcelona, Barcelona, Spain. [8]Department of Biochemistry and Molecular Biology B and Immunology, Faculty of Medicine, University of Murcia, 30120 Murcia, Spain. [9]Present address: Interfaculty Institute for Cell Biology, Department of Immunology, University of Tübingen, Auf der Morgenstelle 15, 72076 Tübingen, Germany. ✉e-mail: pablopel@um.es

