## [Peer Review File · Nature Communications]

Pathogenic NLRP3 mutants forms constitutively active inflammasomes resulting in immune-metabolic limitation of IL-1 β productionREVIEWER COMMENTS

Reviewer #1 (Remarks to the Author):

In their clearly presented manuscript, "The NLRP3 of Cryopyrin-Associated Periodic Syndromes forms a constitutively active inflammasome and results in monocyte immunometabolism alteration.", Molina-Lopez et al. take a methodical and comprehensive approach to study the mechanisms of CAPS mutations. They studied human blood, PBMCs, and isolated monocytes from CAPS patients, immortalized recombinant NLRP3 mouse macrophages, and recombinant HEK cells as well as publicly available transcriptional data, that allowed for delineating pathways and control for varying expression of NLRP3. This work attempts to understand a very complicated and interactive pathway with some previous conflicting data due to different cells, timing and conditions.

The strengths of this paper are

1. Logical evaluation of mechanisms of mutant NLRP3 that explores the role of NLRP3 expression and function in the absence and presence of LPS and other stimuli
2. Use of cells from CAPS patients and controls and several different novel cell model systems , stimuli, and readouts (ie p2x7r shedding)
3. Study of ubiquitination and metabolic reprogramming that has not been adequately addressed in other studies related to NLRP3.
4. Data and interpretation that come together for an understandable story.

The primary weaknesses

1. There are clear and some consistent differences between different mutations in the functional studies and response to MCC950 of the different mutations that are not discussed. In addition there are differences in response to different NFkB activators
2. At times It is a confusing to follow which cell model system is being used for each experiment while reading the results and looking at figures.
3. English writing could be improved in multiple locations throughout the manuscript, especially the abstract where every sentence is awkwardly written.

Comments:

Title: I would change to "Mutant NLRP3 of...alteration in monocyte immunometabolism"

Abstract: Accurate but poorly written

Introduction

Concise and comprehensive and appropriate for this audience
Good review of current state of knowledge of this disease and mechanisms

Line 74 is not clear

Methods

Adequate detail of techniques

Results

Clear presentation of data but many awkwardly written sentences

While figure legends are labeled adequately, in the figures with multiple cell models, it would be helpful to the reader to label the panels with different cell model systems (ie. Fig 1, 3, 4, 6)

While description of transcriptional analysis in Methods are adequate, it would be helpful to provide more information about cell type and conditions in the Results when introducing the data

Fig 1. .

The clear demonstration of different mutations by color is very helpful .Variability in human samples is expected, but the data does show differences, particularly A439T which separates from the other mutations and D303N and T348M which does not separate from Healthy in several conditions. This should be discussed

Appropriate inclusion of reanalysis of publicly available data but label 1F better

Fig 2. I understand the goal of controlling for differing expression levels which is important, but I don't understand the need for using different doxycycline concentrations for WT and Mutant since there is very little GDMD cleavage and likely casp1 cleavage in the WT even at doxy doses equal to the mutant.

If possible, I would want to see co-transfection of mutant and WT as is observed in CAPS patients that are heterozygous and likely have NLRP3 inflammasomes with a mix of mutant and WT in their cells. I realize it may be difficult to account for different efficiency and expression.

Fig 3

3b and c also have different symbols for experiments and should be stated in the legend

Fig 4

The different colors in panels a-c are not labeled well or consistently with other figures or with any clear pattern within this figure. While the differences observed in palmitate and S100A9 are mentioned in general, there may be differences between mutations and with MCC950 the could be mentioned

Fig 5 clear

Fig 6D color may not be necessary. It's a different cell model system but could be labeled in figure

Fig 7

This data may require additional explanation and technique may not be familiar to readers

Fig 8

A,B needs more explanation since technique may not be familiar to readers

Supplementary Figures

Appropriate placement of supplemental data except that S2E is out of order

Discussion

Data supports conclusions

This section has several awkwardly written sentences

References - Adequate

Reviewer #2 (Remarks to the Author):

The current manuscript by Molina-Lopez et al. characterizes the constitutively active NLRP3 variants that are associated with CAPS. The authors show evidence in various model systems that CAPS mutations augment inflammasome activation independent of NFkB-related priming steps. Moreover, the authors identify evidence of immunometabolic programs being differentially regulated in CAPS cells compared to control.

The authors present a convincing set of data that highlights the chronic upregulation of the NLRP3 inflammasome in primary cells of CAPS patients and in a new inducible model of NLRP3 variants in immortalized BMDM. Moreover, the authors make good use of available datasets to support their study. The link to ubiquitination is evident (could also be teased out mechanistically if particular Lys residues were identified).

For most experiments, the authors do well to interpret the data accurately and cautiously and have necessary controls most of the time. The inclusion of NLRP3 inhibition through MCC-950 provides a useful tool to tease out dependent mechanisms. The data linking constitutive activation in the absence of priming is interesting and for the most part, provided with multiple layers of evidence.

I have some constructive thoughts that aim to strengthen the current manuscript.

Firstly, the link to altered metabolism is shown, but the significance of these potential alterations lack clarity and are observational at this point in time. Starting with the metabolomics data, it will be necessary to identify the metabolites in the untargeted screen. I realize that sample size is at a premium when dealing with clinical samples. However, it is uncommon to report metabolites without identification. As presented, the PCA analyses make sense, to verify the similarities between groups. My suggestion would be to (if possible) obtain sufficient sample from clinical patients to identify the metabolites that are differentially present in monocytes (likely something the authors would have done if possible). A close second option would be to perform untargeted metabolomics on the iMos system the authors use throughout and for Figure 8.

Along these lines, the analyses performed to tease out differences in the transcript expressions of PPP and glycolysis related genes are supportive, but not sufficient. The authors have the chance to standardize the measurements across the paper to their inducible system, why not measure targeted transcript expression by qPCR at minimum or perform RNAseq on these cells? Moreover, with identification of metabolites, there would be the ability for the integration of these systems-based approaches that could uncover novel mechanisms.

The Seahorse data is interesting but requires consideration. The data shows that in the iMos system, CAPS variant expression lowers ECAR, which is subsequently "rescued" by MCC-950. The authors speculate that pyroptosis may affect glycolytic enzymes; however, this statement is vague and does not explain the phenotype. The data shown in Figure 8B/C and E seem to represent the technical replicates. Therefore, statistical analysis on this data is misleading and perhaps not possible. Figure 8D is said to be n=4 independent replicates (biological replicates); however, there are only 3 data points. The authors should show the full results of the mito-stress test as both basal and maximal respiration are similar, calling into question the efficacy of the FCCP (assuming FCCP was used to uncouple and determine maximal respiration). More consideration should be made to mechanistically test the reason and consequence of the metabolic shift in these cells.

I understand why the empty vector and WT control conditions were included early (Figures 2-4) and left out for Figures 5-6 (though it would have been nice to have these included). However, it would strengthen the story to include the WT control conditions for all of the experiments in figure 7 (as they are re-imagined) and figure 8 (metabolic profiling). An important consideration is also that these are immortalized cells whose cellular metabolic programs are not guaranteed to be the same as primary BMDM or in vivo macrophages.

The manuscript would benefit from the inclusion of a limitations section within the discussion.

As a minor point, at Line 351- it is unclear as to what is meant by "measurement of IL1B release in CAPS cells cannot be used as a marker of inflammasome

Reviewer #3 (Remarks to the Author):

The Cryopyrin-Associated Periodic Syndrome (CAPS) is an autoinflammatory condition caused by monoallelic variants in the NLRP3 gene that exacerbate IL-1b production. The regulatory mechanisms for the NLRP3 CAPS-inflammasome are not yet well understood. This study found that CAPS-associated human NLRP3 variants, expressed in Nlrp3-deficient mouse macrophages, or when investigated in patient characterized CAPS monocytes, result in constitutively active NLRP3 inflammasomes, inducing a basal cleavage of gasdermin D, IL-18 release, and pyroptosis. Importantly, this has been shown to occur in the absence of any external triggers or NF-κB-dependent priming. Furthermore, as expected, priming of NLRP3 CAPS-inflammasomes resulted in secretion of IL-1b, HMGB1, and other associated molecules. Finally, the authors attempt to associate these findings with impaired glycolysis in NLRP3 CAPS-inflammasome carrying monocytes.

The research is highly relevant and could have implications for the treatment of NLRP3-dependent inflammatory diseases. However, the findings lack much novelty and are only incremental in nature with much of the findings either expected or described previously.

1. The whole premise of this research is based on the fact that inflammasome activation is independent of the priming event. However, this has previously been demonstrated by several studies both in mouse macrophages and in human monocytes. Moreover, data in Fig. 5 demonstrates that the constitutive activity of the CAPS inflammasome is modulated by NF- κ B signalling.

2. The dependence on DUB of the CAPS inflammasome is again only modest in terms of the effect on IL-18 (Fig. 6). Prior priming of the CAPS inflammasome did not inhibit IL-18 to statistically significant levels in the presence of DUB inhibitors (Fig. 6f), though priming resulted in significantly increased IL-18 levels in monocytes from CAPS patients (Fig. 1c), and DUB inhibitors decrease ASC specks in primed CAPS inflammasomes (Fig. 6h).

3. The metabolic data at best shows only an association between impaired glycolysis and the CAPS inflammasome. The analysis of GEO datasets sheds some light on the mechanistic details but whether these are the cause or the consequences during CAPS inflammasome remains unknown. Anakinra blocks far downstream in the process than the focus of this research so is not much useful here.

4. Line 112 - The data shown until this point does not necessarily implicate a role for NF- κ B activation in enhanced ASC specks observed in CAPS monocytes particularly when the role of LPS in post-translational modifications has been highlighted in the introduction.

5. If the healthy monocytes show increased ASC specks upon LPS priming in Fig. 1a, why is there no corresponding increase in IL-18 or IL-1 β (4h in 1d)? This needs to be explained along with the second stimuli that the control healthy monocytes here are responding to?

6. Fig. 2B is missing b-actin control expression.

7. The manuscript contains minor grammatical errors and can be edited by a native speaker.

Point-by-point response to reviewers' comments:

MS No.: NCOMMS-23-19097

MS Title: Mutant NLRP3 causing Cryopyrin-Associated Periodic Syndromes forms a constitutively active inflammasome and results in alteration in monocyte immunometabolism to limit IL-1 β production.

General: We would like to express our gratitude to the reviewers for their insightful feedback, which has significantly enhanced the manuscript's quality. We have conducted further experiments and made revisions as per their recommendations. The revised manuscript addresses all their comments. Changes made are highlighted using the tracking tool in Word.

Reviewer #1:

In their clearly presented manuscript, "The NLRP3 of Cryopyrin-Associated Periodic Syndromes forms a constitutively active inflammasome and results in monocyte immunometabolism alteration.", Molina-Lopez et al. take a methodical and comprehensive approach to study the mechanisms of CAPS mutations. They studied human blood, PBMCs, and isolated monocytes from CAPS patients, immortalized recombinant NLRP3 mouse macrophages, and recombinant HEK cells as well as publicly available transcriptional data, that allowed for delineating pathways and control for varying expression of NLRP3. This work attempts to understand a very complicated and interactive pathway with some previous conflicting data due to different cells, timing and conditions.

The strengths of this paper are:

- 1. Logical evaluation of mechanisms of mutant NLRP3 that explores the role of NLRP3 expression and function in the absence and presence of LPS and other stimuli*
- 2. Use of cells from CAPS patients and controls and several different novel cell model systems, stimuli, and readouts (ie p2x7r shedding)*
- 3. Study of ubiquitination and metabolic reprogramming that has not been adequately addressed in other studies related to NLRP3.*
- 4. Data and interpretation that come together for an understandable story.*

Answer: We value the reviewer' feedback highlighting the strengths of our study.

The primary weaknesses:

- 1. There are clear and some consistent differences between different mutations in the functional studies and response to MCC950 of the different mutations that are not discussed. In addition there are differences in response to different NF κ B activators*

Answer: We thank the reviewer for highlighting this crucial aspect. We concur that certain mutations in NLRP3 exhibit consistent differences in MCC950 response and NF- κ B activators compared to other mutations. These issues have now been addressed in our revised manuscript:

Page 7: "However, MCC950 presented a less potent inhibitory effect on the NLRP3 p.T348M variant, affecting both GSDMD processing and IL-18 release (**Figure 2A,E**). This observation

aligns with a prior report indicating higher IC₅₀ for MCC950 in PBMCs from CAPS patients carrying the p.T348M variant compared to the p.A439V and p.E311K variants⁹. Furthermore, in recombinant systems, MCC950 was unable to completely eliminate cells with NLRP3 p.T348M-associated puncta²². These findings suggest that therapy strategies involving MCC950 analogues might require dosage adjustments based on the specific *NLRP3* variant carried by patients”.

Page 8/9: “...other NF-κB activators, such as palmitate, S100A9 or IL-6 acted as inducers of NLRP3-dependent IL-1β release when pathogenic NLRP3 variants p.R260W, p.D303N and p.T348M were expressed, which was not the case with the expression of the wild type allele (**Figure 4A, S3A-C**)”.

Page 9: “Notably, the p.T348M NLRP3 variant exhibited a subdued IL-1β release compared to the p.R260W and p.D303N NLRP3 variants (**Figure 4A**). Both palmitate and IL-6 were more effective in inducing IL-1β release when the p.D303N NLRP3 variant was expressed (**Figure 4A**)”.

Page 19: “Interestingly, responses varied among the different NLRP3 mutations we studied; the p.T348M mutation, in particular, resulted in reduced IL-1β release”.

2. At times It is a confusing to follow which cell model system is being used for each experiment while reading the results and looking at figures.

Answer: We have now added the cell model used in each figure panel for clarity.

3. English writing could be improved in multiple locations throughout the manuscript, especially the abstract where every sentence is awkwardly written.

Answer: We have now improved English style and grammar.

Comments:

Title: I would change to “Mutant NLRP3 of...alteration in monocyte immunometabolism”

Answer: We have now changed and improved the title as suggested by the reviewer.

Abstract: Accurate but poorly written

Answer: We have now improved English style and grammar. Special attention was on the sentences of the Abstract.

Introduction

Concise and comprehensive and appropriate for this audience

Good review of current state of knowledge of this disease and mechanisms

Answer: We appreciate the reviewer comments on the introduction.

Line 74 is not clear

Answer: We have clarified the sentence:

Page 3: “While specific *NLRP3* variants have been associated with each CAPS clinical phenotype, current understanding suggests that some *NLRP3* variants may cause overlapping phenotypes. This supports the idea that CAPS behaves more like a syndrome with a variable spectrum of features rather than a singular entity⁵”.

Methods

Adequate detail of techniques

Answer: We appreciate the reviewer comments on the methods.

Results

Clear presentation of data but many awkwardly written sentences

Answer: We have now improved English style and grammar. We hope now the sentences are improved.

While figure legends are labeled adequately, in the figures with multiple cell models, it would be helpful to the reader to label the panels with different cell model systems (ie. Fig 1, 3, 4, 6)

Answer: As requested, we have now added the cell model used in each figure panel for clarity.

While description of transcriptional analysis in Methods are adequate, it would be helpful to provide more information about cell type and conditions in the Results when introducing the data

Answer: In the Results, when the transcriptional analysis of GEO datasets are mentioned, the cell type and treatments are now mentioned in the text (see underline):

Page 6: “To investigate the effect of LPS stimulation on human monocyte activation, we analysed the GEO dataset GSE42606. The analysis revealed that LPS upregulates the

expression of *IL1B*, *TNFA*, and *NLRP3* in healthy human PBMCs, while the expression of *IL18* and *LGALS3* remains unchanged (**Figure 1F**). Contrarily, in active CAPS patients carrying the p.G569R *NLRP3* variant, no discernible alteration in the expression patterns of *IL1B*, *IL18* and *NLRP3* in blood cells was observed, according to the analysis of GEO dataset GSE57253 (**Figure 1G**)”

Page 9: “Further examination of the GEO dataset GSE57253 revealed increased *S100A9* gene expression in blood cells from active CAPS patients when compared to healthy donors (**Figure 4G**). Remarkably, anakinra treatment in CAPS patients reduced *S100A9* gene expression (**Figure 4G**)”.

Page 14: “We then proceeded to examine the expression of metabolic-associated gene expression in two independent GEO datasets (GSE57253 and GSE17732) that included blood samples obtained from CAPS patients during an active disease periods^{30,31}. Interestingly, our analysis revealed a marked differential expression of genes associated with glycolysis and the pentose phosphate pathway in blood cells of active CAPS patients. Moreover, upon patients treatment with IL-1 inhibitor anakinra, the expression of these genes appear to normalize (**Figure 7D, S7A**). Genes associated with pentoses phosphate (*DERA*, *PGM2*, *NAMPT*) and glycolysis (*BPGM*, *LDHA*, *PFKFB3*) revealed elevated expression in blood cells from active CAPS patients, and a decrease following anakinra treatment (**Figure 7D, S7A**)”.

Fig 1. .

The clear demonstration of different mutations by color is very helpful .Variability in human samples is expected, but the data does show differences, particularly A439T which separates from the other mutations and D303N and T348M which does not separate from Healthy in several conditions. This should be discussed

Answer: We agree with the reviewer. Indeed, for certain markers such as galectin 3 or IL-1 β , there was a distinct separation in the response of samples from patients carrying the p.A439T mutation. However, this difference was not observed for other markers, such as ASC specks, which serve as an indicator of inflammasome oligomerization. As suggested, we have now addressed this point in the revised manuscript:

Page 5/6: “We also observed a genotype-dependent differential release of galectin-3 and IL-1 β , being higher in patients carrying the p.A439T variant compared to those carrying the remaining *NLRP3* variants (**Figure 1B,D**). Nevertheless, the observed increase in the percentage of ASC-specking monocytes was consistent across all analysed variants (**Figure 1A**). This observation raises an intriguing hypothesis, while different *NLRP3* variants may prompt constitutive inflammasome assembly, subsequent downstream signalling could be distinctly regulated”.

Appropriate inclusion of reanalysis of publicly available data but label 1F better

Answer: As requested, we have now added the GEO accession number in panels of the Figure 1F and 1G for clarity. We also added these accession numbers in the other figures where GEO databases are used.

Fig 2. I understand the goal of controlling for differing expression levels which is important, but I don't understand the need for using different doxycycline concentrations for WT and Mutant since there is very little GDMD cleavage and likely casp1 cleavage in the WT even at doxy doses equal to the mutant.

Answer: The primary goal was to ensure comparable expression levels of both the WT and mutant NLRP3. This was accomplished by using varying doses of doxycycline, thereby ensuring that any observed differences in the results are attributable to the mutation rather than potential disparities in expression. As the reviewer rightly points out, even at high doses of doxycycline—which induce higher levels of WT NLRP3 expression compared to the mutants—there were no indications of caspase-1 activation. A dose-response relationship for doxycycline concentrations is depicted in Figure 2A and E. Consequently, we believe that when comparing WT and mutant NLRP3 in terms of the number of ASC speck-forming macrophages and cell death (as shown in panels 2C and 2D), it is more crucial to maintain consistent NLRP3 expression levels rather than using the same dose of doxycycline as an inducer of NLRP3 expression.

If possible, I would want to see co-transfection of mutant and WT as is observed in CAPS patients that are heterozygous and likely have NLRP3 inflammasomes with a mix of mutant and WT in their cells. I realize it may be difficult to account for different efficiency and expression.

Answer: We have now performed the recommended experiment, co-transfecting both the mutant and WT NLRP3 into the same cell to simulate the heterozygous genotype found in CAPS patients. In the updated **Supplementary Figure S2E**, we present a control Western blot to analyze the expression of NLRP3 WT and NLRP3-D303N-YFP in co-transfections. This approach allowed us to differentiate between WT and mutant NLRP3 in terms of size on the same blot, using the same anti-NLRP3 antibody. Under these conditions we measured ASC specking cells by flow cytometry. As shown in the revised **Figure 3D**, the co-expression of WT and mutant NLRP3 resulted in a higher percentage of cells with ASC specks compared to when the mutant NLRP3 is expressed alone. This difference was significant for low NLRP3 expression, suggesting that WT NLRP3 could enhance the basal activity of the NLRP3 p.D303N inflammasome when this receptor is expressed at low concentrations.

We revised accordingly the MS in page 8: “In an attempt to mimic the typical heterozygous genotype of CAPS patients, we co-expressed both wild type and p.D303N NLRP3 in the same cells (**Figure S2E**). Data showed that the wild type NLRP3 induced a significant increase in cells

with ASC oligomers at low p.D303N NLRP3 expression levels, but not at high levels (**Figure 3D**). This suggests that even low concentrations of mutant NLRP3 can enable the wild type to potentially intensify the basal autoactivation of the mutant NLRP3 inflammasome”.

Fig 3

3b and c also have different symbols for experiments and should be stated in the legend

Answer: We have now modified the figure legend to explain that: “each independent experiment is represented by a different symbol in the histograms”.

Fig 4

The different colors in panels a-c are not labeled well or consistently with other figures or with any clear pattern within this figure. While the differences observed in palmitate and S100A9 are mentioned in general, there may be differences between mutations and with MCC950 the could be mentioned

Answer: We have now removed colors from panels A-C in Figure 4. In this panel MCC950 was only used on p.D303N NLRP3 mutation. However, we have now mentioned differences on MCC950 effect related to the different mutations in the revised MS:

Page 7: “However, MCC950 presented a less potent inhibitory effect on the NLRP3 p.T348M variant, affecting both GSDMD processing and IL-18 release (**Figure 2A,E**). This observation aligns with a prior report indicating higher IC₅₀ for MCC950 in PBMCs from CAPS patients carrying the p.T348M variant compared to the p.A439V and p.E311K variants⁹. Furthermore, in recombinant systems, MCC950 was unable to completely eliminate cells with NLRP3 p.T348M-associated puncta²². These findings suggest that therapy strategies involving MCC950 analogues might require dosage adjustments based on the specific *NLRP3* variant carried by patients”.

Fig 5 clear

Fig 6D color may not be necessary. It's a different cell model system but could be labeled in figure

Answer: As suggested, we have now removed color from panel D in Figure 6 and we have labeled the cell types used in the different Figure panels for clarity.

Fig 7

This data may require additional explanation and technique may not be familiar to readers

Answer: The technique procedure is now explained in the methods section avoiding acronyms, and we have now added detailed explanation for the results of this Figure:

Page 12: “The pre-processing of the entire metabolite dataset resulted in a data matrix comprising 203 metabolites in negative polarity and 925 in positive polarity, from which they were reduced to 603 metabolites in positive polarity after removal metabolites with high variability among replicates (**Table S2**). To efficiently represent the variability of these metabolites among monocytes of healthy donors and CAPS patients while preserving trends and patterns, we employed a principal component analysis (PCA) model on the final merged data matrix (**Figure 7A**). The calculated PCA model was built based on 16 samples and three components, with the first two principal components (PC1 and PC2) accounting for 25.2% and 9.6% of the overall variability, respectively. PCA model results illustrated variations based on the total covariance among the study cohorts. We observed that the metabolomic variation of the untreated monocytes from CAPS patients clustered more closely with the LPS-treated monocytes from healthy individuals (**Figure 7A**), suggesting an inherent pro-inflammatory profile of CAPS monocytes”.

Fig 8

A,B needs more explanation since technique may not be familiar to readers

Answer: We have now added additional explanation for the technique using in this Figure:

Page 16: “To further investigate whether alterations in gene expression led to functional changes in macrophages expressing either mutant or wild-type NLRP3, we examined glycolysis functionality. To achieve this, we evaluated glycolysis rate by measuring the acidification of extracellular media produced by mitochondria in response to proton efflux and measured the oxygen consumption rate of these macrophages”.

Supplementary Figures

Appropriate placement of supplemental data except that S2E is out of order

Answer: We appreciate reviewer suggestion, previous Figure S2E is now Figure S3D to follow the order the figures as they appear in the text.

Discussion

Data supports conclusions

This section has several awkwardly written sentences

Answer: We appreciate the comment about the robustness of our conclusions. We have now corrected the text for English style and grammar. Now we hope the sentences in the discussion are improved.

References - Adequate

Answer: We have however included additional references to answer specific points raised by the reviewers.

Reviewer #2:

The current manuscript by Molina-Lopez et al. characterizes the constitutively active NLRP3 variants that are associated with CAPS. The authors show evidence in various model systems that CAPS mutations augment inflammasome activation independent of NFkB-related priming steps. Moreover, the authors identify evidence of immunometabolic programs being differentially regulated in CAPS cells compared to control.

The authors present a convincing set of data that highlights the chronic upregulation of the NLRP3 inflammasome in primary cells of CAPS patients and in a new inducible model of NLRP3 variants in immortalized BMDM. Moreover, the authors make good use of available datasets to support their study. The link to ubiquitination is evident (could also be teased out mechanistically if particular Lys residues were identified).

For most experiments, the authors do well to interpret the data accurately and cautiously and have necessary controls most of the time. The inclusion of NLRP3 inhibition through MCC-950 provides a useful tool to tease out dependent mechanisms. The data linking constitutive activation in the absence of priming is interesting and for the most part, provided with multiple layers of evidence.

Answer: We appreciate the comments of the reviewer.

I have some constructive thoughts that aim to strengthen the current manuscript.

Firstly, the link to altered metabolism is shown, but the significance of these potential alterations lack clarity and are observational at this point in time. Starting with the metabolomics data, it will be necessary to identify the metabolites in the untargeted screen. I realize that sample size is at a premium when dealing with clinical samples. However, it is uncommon to report metabolites without identification. As presented, the PCA analyses make sense, to verify the similarities between groups. My suggestion would be to (if possible) obtain sufficient sample from clinical patients to identify the metabolites that are differentially present in monocytes (likely something the authors would have done if possible). A close second option would be to perform untargeted metabolomics on the iMos system the authors use throughout and for Figure 8.

Answer: We have now presented a tentative identification of metabolites using their exact molecular mass and molecular formula. We were unable to confirm the metabolites by targeted metabolomics due to the small quantity of the available sample from CAPS patients. For a list of identified metabolites, please see new **Supplementary Table S3** and **Supplementary Figure S6B**.

As recommended, we carried out untargeted metabolomics on immortalized macrophages, both expressing and not expressing NLRP3 p.D303N. We obtained metabolomic data from *Nlrp3*^{-/-} immortalized macrophages treated with doxycycline and compared to *Nlrp3*^{-/-} immortalized macrophages treated with doxycycline to induce expression of NLRP3 p.D303N. Both sets were either untreated or treated with LPS. The PCA analysis (see new **Supplementary Figure S6C**) revealed a metabolic distribution similar to that found in human monocytes from healthy individuals and CAPS patients. In immortalized macrophages the expression of NLRP3 p.D303N resulted in a metabolic landscape resembling that of LPS-treated macrophages without NLRP3 expression. From that dataset, we also performed a tentative identification of metabolites using their exact molecular mass and molecular formula (see new **Supplementary Table S4**).

From our tentative identification of metabolites, we found that both CAPS patient samples and immortalized macrophages with the CAPS NLRP3 variant exhibited alterations in similar metabolites (see new **Supplementary Figure S6B**, and **Supplementary Tables S3 and S4**). These primarily included metabolites related to amino acid, lipids and lipid-like molecules.

Furthermore, we analyzed glycolysis-related metabolites (lactate and pyruvic acid) as well as metabolites of the TCA cycle in immortalized macrophages expressing either wild type or p.D303N NLRP3. We discovered that the expression of mutant NLRP3 led to a decrease in glycolysis metabolites. See new data in the **Supplementary Figures S8A,C**.

Along these lines, the analyses performed to tease out differences in the transcript expressions of PPP and glycolysis related genes are supportive, but not sufficient. The authors have the chance to standardize the measurements across the paper to their inducible system, why not measure targeted transcript expression by qPCR at minimum or perform RNAseq on these cells? Moreover, with identification of metabolites, there would be the ability for the integration of these systems-based approaches that could uncover novel mechanisms.

Answer: As recommended by the reviewer, we have now conducted RNAseq on immortalized macrophages expressing either the wild type or the p.D303N NLRP3 variant, both in the presence and absence of MCC950. We discovered that the basal expression of the p.D303N NLRP3 variant led to a specific decrease in the expression of glycolytic genes. This expression was restored when MCC950 was used (see new **Figure 8A and B**). These changes were not observed when the wild type NLRP3 was expressed (see new **Figure 8B**).

We have now identified the main pathways affected by the constitutive activation of the NLRP3 p.D303N inflammasome, described by the upregulation or downregulation of specific genes (see new **Figure 8C**, **Tables S6, S7**). We restricted our analysis to include genes that were

upregulated following NLRP3 p.D303N expression and subsequently downregulated with MCC950 treatment. In particular, the expression of NLRP3 with the p.D303N variant led to a specific upregulation in genes associated with the inflammatory response. This includes the induction of nuclear receptors 4A1 and 4A2, which initiate a pro-inflammatory program in macrophages, although these receptors have not been previously linked to CAPS. The expression of NLRP3 p.D303N also tended to enhance pathways related to viral response (see new **Figure 8C, Tables S6**), as well as genes associated with pentose phosphate pathway, amino acid pathways and lipid metabolism (which supports the metabolomic data, see new **Figure 8C, S7B, S8D**). These changes were decreased when the NLRP3 activity was inhibited by MCC950 (see new **Figure 8C, S7B, S8D**). Additionally, our RNAseq on immortalized macrophages is now compared to our previous analysis of GEO databases in human CAPS samples (see **Figure 7D,E**).

The seahorse data is interesting but requires consideration. The data shows that in the iMos system, CAPS variant expression lowers ECAR, which is subsequently “rescued” by MCC-950. The authors speculate that pyroptosis may affect glycolytic enzymes; however, this statement is vague and does not explain the phenotype.

Answer: We now present targeted metabolic data (refer to new **Figure S8A,C**) and RNAseq data (see new **Figures 7E, 8, S7B, S8D**) that demonstrate a decrease in glycolysis metabolites and glycolytic genes downregulation when the NLRP3 p.D303N variant is expressed, but not when the wild type NLRP3 is expressed. This decrease can be attributed to the basal activity of the CAPS-associated NLRP3 inflammasome, since the incubation with MCC950 restored the expression of glycolytic genes and metabolites (see new **Figures 7E, 8A,B**). It is important to highlight that MCC950 had no effect when the wild type NLRP3 was expressed (see new **Figure 8B**). We hope now we provide a more robust set of evidence indicating a decrease of glycolysis in CAPS myeloid cells.

The data shown in Figure 8B/C and E seem to represent the technical replicates. Therefore, statistical analysis on this data is misleading and perhaps not possible.

Answer: The data presented in the previous Figure 8B,C,E (now **Figure 9B,C,E**) are derived from the average of various biological replicates, consisting of 3 to 8 different cell culture wells prepared on the same day. These averages then represent 3 to 4 independent experiments from different days. Given that the averages were calculated from multiple cell cultures ($N \geq 3$), we conducted statistical analysis on these data. For each independent experiment conducted on different days, we consistently observed statistically significant difference. This approach is now detailed in the figure legend.

Figure 8D is said to be n=4 independent replicates (biological replicates); however, there are only 3 data points.

Answer: We believe the reviewer is referring to Figure 8C (now **Figure 9C**), not 8D. We apologize for the error in the original legend, which stated that the figure panel was based on $N=4$ data points, when it was actually performed with $N=3$ data points. This has now been corrected in the figure legend. As for Figure 8D (now **Figure 9D**), it is derived from $N=3-8$

biological replicates. However, individual data points are not represented due to the overlap of two histograms, which would complicate the representation of overlapping data points. For individual values for this panel, please refer to the uploaded raw data files.

The authors should show the full results of the mito-stress test as both basal and maximal respiration are similar, calling into question the efficacy of the FCCP (assuming FCCP was used to uncouple and determine maximal respiration).

Answer: As requested, the complete profile of the mito-stress test is now showed in the newly added **Supplementary Figure S8B**. The Cell Mito-stress kit from Agilent uses FCCP to uncouple mitochondrial respiration. Our initial optimization of the technique in immortalized macrophages revealed that FCCP at concentrations above 0.5 μM were actually reducing maximal respiration, likely due to toxic effects:

Consequently, we used a concentration of 0.5 μM FCCP for the subsequent mito-stress test. However, at this concentration, FCCP was either unable to increase the maximal respiration beyond basal respiration, or the macrophages were already at maximal respiration rate at basal conditions. This is a common phenomenon observed in different cell types, as confirmed by Agilent's technical department, and is a limitation of this assay. Nonetheless, the full profiles of the mito-stress test are now presented, and no significant changes in mitochondrial respiration were observed when the NLRP3 p.D303N variant was expressed (see new **Supplementary Figure S8B**). This aligns with the new data obtained for TCA metabolites and TCA-related gene expression in these cells, indicating that the presence of NLRP3 p.D303N does not affect this metabolic pathway (see new **Supplementary Figure S8C,D**).

More consideration should be made to mechanistically test the reason and consequence of the metabolic shift in these cells.

Answer: To better understand the functional implications of glycolysis impairment on the inflammatory response of CAPS, we conducted a series of new experiments. These experiments utilized either pyruvate to circumvent the reduction in glycolysis caused by the NLRP3 p.D303N, or 2DG to inhibit the activity of the glycolytic enzyme hexokinase, thereby further reducing glycolysis.

Our findings revealed that supplementing the cell culture media with pyruvate increased the basal release of IL-1 β , but not IL-18, in unprimed macrophages expressing the NLRP3 p.D303N variant. This effect was not observed in macrophages expressing the wild type NLRP3 (see new **Figure 10A**). Conversely, we found that 2DG reduced IL-1 β release in

macrophages expressing the p.D303N NLRP3 variant (see new **Figure 10C**). However, IL-18 release was not affected by 2DG (see new **Figure 10C**).

These results suggest that the decrease in glycolysis found under basal conditions of NLRP3 p.D303N could limit IL-1 β production (thereby impairing inflammatory flares) but would not impact on IL-18 production or in a basal inflammatory state important for protection against pathogens. Interestingly, a slight increase in IL-1 β gene expression was observed when the p.D303N variant, but not the wild type NLRP3, was expressed in macrophages. This increase was reduced by MCC950 (see new **Figure 10B**).

This minor upregulation of the IL-1 β gene is part of the pro-inflammatory phenotype we observed following the expression of NLRP3 with CAPS-associated variants. The impairment of glycolysis (due to downregulation of glycolytic gene expression upon constitutive formation of a NLRP3 inflammasome with CAPS variants) could play a crucial role in reducing the basal maturation and release of IL-1 β , and therefore limiting pathological inflammatory flares.

I understand why the empty vector and WT control conditions were included early (Figures 2-4) and left out for Figures 5-6 (though it would have been nice to have these included). However, it would strengthen the story to include the WT control conditions for all of the experiments in figure 7 (as they are re-imagined) and figure 8 (metabolic profiling).

Answer: We have now incorporated new experiments involving immortalized macrophages with both the empty vector and the wild type NLRP3 expression. These are included in what was previously Figure 8 (now **Figure 9**), as well as in the new RNAseq data (presented in the new **Figure 8**). As for Figure 7, data from CAPS patients is consistently compared with that of healthy donors.

An important consideration is also that these are immortalized cells whose cellular metabolic programs are not guaranteed to be the same as primary BMDM or in vivo macrophages.

Answer: We agree with the reviewer that the metabolic profile of immortalized macrophages may differ from those of primary BMDM, human monocytes, or even human macrophages, both *in vitro* and *in vivo*. This is a common constraint in all studies utilizing cellular models. Nevertheless, we have observed a similar metabolic profile and gene expression when using monocytes from CAPS patients and immortalized macrophages expressing the NLRP3 p.D303N variant (**Figures 7,S6,S7**). We have now incorporated in the Discussion a paragraph about the limitations of our study:

Page 22: "One limitation of this study is the potential discrepancy between the gene expression and metabolic programs between immortalized mouse macrophages expressing the p.D303N NLRP3 variant and those from primary mouse macrophages or human monocytes/macrophages, both *in vitro* and *in vivo*. Furthermore, our study solely focused on the metabolism affected by one pathogenic NLRP3 variant (p.D303N) expressed in immortalized macrophages. Despite this, we observed that the metabolic profile and gene expression in monocytes from human CAPS patients carrying additional pathogenic NLRP3

variants were similar to those in immortalized macrophages expressing the p.D303N human NLRP3 variant. This observation suggests that a reduction in glycolysis could limit IL-1 β -derived inflammatory flares in CAPS”.

The manuscript would benefit from the inclusion of a limitations section within the discussion.

Answer: As explained above, we have included the limitations of the study in the discussion.

As a minor point, at Line 351- it is unclear as to what is meant by “measurement of IL1B release in CAPS cells cannot be used as a marker of inflammasome

Answer: In the text, we highlight that the absence of NF- κ B activation (i.e., without LPS treatment) precludes the use of IL-1 β release a single marker for inflammasome activation. Under these conditions, myeloid cells do not produce pro-IL-1 β or pro-IL-1 β levels are very low, and even if inflammasomes are active, both ELISA and Western blot tests will yield negative results for the release of IL-1 β . Various studies measure IL-1 β release as the sole marker of inflammasome activation. We aim to emphasize the importance of assessing multiple inflammasome-related markers, particularly in resting or non-primed conditions. We have now rephrased the text to make it clear:

Page 19: “Consequently, the measurement of IL-1 β release in CAPS-related cells should not be the only marker of inflammasome activation in the absence of NF- κ B induction”.

Reviewer #3:

The Cryopyrin-Associated Periodic Syndrome (CAPS) is an autoinflammatory condition caused by monoallelic variants in the NLRP3 gene that exacerbate IL-1b production. The regulatory mechanisms for the NLRP3 CAPS-inflammasome are not yet well understood. This study found that CAPS-associated human NLRP3 variants, expressed in Nlrp3-deficient mouse macrophages, or when investigated in patient characterized CAPS monocytes, result in constitutively active NLRP3 inflammasomes, inducing a basal cleavage of gasdermin D, IL-18 release, and pyroptosis. Importantly, this has been shown to occur in the absence of any external triggers or NF- κ B-dependent priming. Furthermore, as expected, priming of NLRP3 CAPS-inflammasomes resulted in secretion of IL-1b, HMGB1, and other associated molecules. Finally, the authors attempt to associate these findings with impaired glycolysis in NLRP3 CAPS-inflammasome carrying monocytes.

The research is highly relevant and could have implications for the treatment of NLRP3-dependent inflammatory diseases. However, the findings lack much novelty and are only incremental in nature with much of the findings either expected or described previously.

Answer: We appreciate reviewer comment.

1. The whole premise of this research is based on the fact that inflammasome activation is independent of the priming event. However, this has previously been demonstrated by several studies both in mouse macrophages and in human monocytes. Moreover, data in Fig. 5 demonstrates that the constitutive activity of the CAPS inflammasome is modulated by NF- κ B signalling.

Answer: We recognize prior studies that demonstrate the potential for NF- κ B to modulate the NLRP3 inflammasome in CAPS (i.e. references #10 to 16). However, the basal activation of the CAPS NLRP3 inflammasome, a fact that has been overlooked and misunderstood, has not been experimentally investigated until now. Our study aims to fill this gap and therefore we clarified this point throughout the corrected manuscript.

2. The dependence on DUB of the CAPS inflammasome is again only modest in terms of the effect on IL-18 (Fig. 6). Prior priming of the CAPS inflammasome did not inhibit IL-18 to statistically significant levels in the presence of DUB inhibitors (Fig. 6f), though priming resulted in significantly increased IL-18 levels in monocytes from CAPS patients (Fig. 1c), and DUB inhibitors decrease ASC specks in primed CAPS inflammasomes (Fig. 6h).

Answer: We now provide evidence that in CAPS, the control of IL-18 and IL-1 β production can be differentially controlled. While the production of IL-1 β is impacted by a decrease in glycolysis, IL-18 is not (see new **Figure 10**, and detailed explanation in the next point). This may also be reflected in the dependency on DUBs, particularly when LPS is used to prime the cells. The effect of DUB inhibitors on the release of IL-1 β is more pronounced (**Figure 6G**) than for IL-18 (**Figure 6F**) following LPS treatment. Our findings align with a recent study (PMID: 33931568, reference #45) suggesting that DUB inhibitors could be potential treatments for CAPS flares *in vivo*. Consequently, our study further demonstrates that deubiquitination could potentiate the basal activation of the inflammasome in CAPS.

3. The metabolic data at best shows only an association between impaired glycolysis and the CAPS inflammasome. The analysis of GEO datasets sheds some light on the mechanistic details but whether these are the cause or the consequences during CAPS inflammasome remains unknown. Anakinra blocks far downstream in the process than the focus of this research so is not much useful here.

Answer: To better understand the functional implications of glycolysis impairment on the inflammatory response of CAPS, we conducted a series of new experiments. These experiments utilized either pyruvate to circumvent the reduction in glycolysis caused by the NLRP3 p.D303N, or 2DG to inhibit the activity of the glycolytic enzyme hexokinase, thereby further reducing glycolysis.

Our findings revealed that supplementing the cell culture media with pyruvate increased the basal release of IL-1 β , but not IL-18, in unprimed macrophages expressing the NLRP3 p.D303N variant. This effect was not observed in macrophages expressing the wild type

NLRP3 (see new **Figure 10A**). Conversely, we found that 2DG reduced IL-1 β release in macrophages expressing the p.D303N NLRP3 variant (see new **Figure 10C**). However, IL-18 release was not affected by 2DG (see new **Figure 10C**).

These results suggest that the decrease in glycolysis found under basal conditions of NLRP3 p.D303N could limit IL-1 β production (thereby impairing inflammatory flares) but would not impact on IL-18 production or in a basal inflammatory state important for protection against pathogens. Interestingly, a slight increase in IL-1 β gene expression was observed when the p.D303N variant, but not the wild type NLRP3, was expressed in macrophages. This increase was reduced by MCC950 (see new **Figure 10B**).

While we agree with the reviewer that Anakinra blocks downstream IL-1 signaling, the impairment of glycolysis found in our study was a direct consequence of the active NLRP3 CAPS-associated inflammasome, as was reverted by the use of MCC950, at the level of metabolites (see new **Figure 9C,D**), glycolytic gene expression (see new **Figures 7E and 8A,B**) and glycolysis function (see **Figure 9A,B**, and new **Figure S8A**). IL-1 β was not directly involved in glycolysis impairment, as the use of IL-1Ra was not affecting glycolysis (new **Supplemental Figure S7C**). However, in patients, we have found in a recent study under review (see attached in confidence) that the canonical NLRP3 inflammasome activation in primary monocytes of a CMML patient after treatment with Anakinra was impaired. This suggests that Anakinra treatment is not only dampening IL-1 signaling, but also is attenuating NLRP3 inflammasome activation (probably affecting IL-1–induced NLRP3 priming in patients).

4. Line 112 - The data shown until this point does not necessarily implicate a role for NF- κ B activation in enhanced ASC specks observed in CAPS monocytes particularly when the role of LPS in post-translational modifications has been highlighted in the introduction.

Answer: We agree with the reviewer, and we have removed the potential role for NF- κ B in this sentence. The sentence now reads:

Page 5: “These results suggest that monocytes from CAPS patients exhibit constitutive NLRP3-inflammasome activation”.

5. If the healthy monocytes show increased ASC specks upon LPS priming in Fig. 1a, why is there no corresponding increase in IL-18 or IL-1 β (4h in 1d)? This needs to be explained along with the second stimuli that the control healthy monocytes here are responding to?

Answer: The reviewer is right, in monocytes from healthy individuals, we observed a statistically significant increase in the release of both IL-18 and IL-1 β after LPS treatment. Specifically for IL-18, the concentration increased from 0.78 \pm 0.13 pg/ml at resting (6h) to 2.18 \pm 0.52 pg/ml after 6h of LPS treatment (**Figure 1C**, refer to raw data file), with a *t*-test *p*-value of 0.0296. For IL-1 β , the concentration increased from 0.65 \pm 0.28 pg/ml at resting (4h) to 11.09 \pm 2.57 pg/ml after 4h of LPS treatment (**Figure 1D**, refer to raw data file), with a *t*-test *p*-value of 0.0037.

However, due to the scale of the graph, this increase may not be immediately apparent. This trend aligns with the observed increase in monocytes with ASC specks following LPS priming, as pointed out by the reviewer. After 4h, the percentage of monocytes with ASC specks increased from $1.92\pm 0.21\%$ at resting to $4.21\pm 0.52\%$ with LPS (**Figure 1A**, refer to raw data file), with a *t*-test *p*-value of 0.0023. After 6 hours, it increased from $4.20\pm 1.31\%$ at resting to $9.74\pm 1.39\%$ with LPS (**Figure 1A**, refer to raw data file), with a *t*-test *p*-value of 0.0071.

We have now explained this in the revised text:

Page 5: "We also found that LPS slightly, but significantly, increased the release of both IL-1 β and IL-18 from the PBMCs of healthy individuals: IL-18 concentration increased from 0.78 ± 0.13 pg/ml at 6h resting to 2.18 ± 0.52 pg/ml after 6h of LPS treatment, with a *t*-test *p*-value of 0.0296; and IL-1 β concentration increased from 0.65 ± 0.28 pg/ml at 4h resting to 11.09 ± 2.57 pg/ml after 4h of LPS treatment, with a *t*-test *p*-value of 0.0037 (**Figure 1C,D**). However, this increase was less pronounced than that observed in the PBMCs of CAPS patients. This corresponds to the observed increase in ASC specks following LPS incubation in monocytes from healthy donors (**Figure 1A**)".

6. *Fig. 2B is missing b-actin control expression.*

Answer: We now incorporated a β -actin loading control for the cell extracts in the revised Figure 2B, which was not previously included as we only displayed supernatants.

7. *The manuscript contains minor grammatical errors and can be edited by a native speaker.*

Answer: We have now corrected the text for English style and grammar.

REVIEWERS' COMMENTS

Reviewer #1 (Remarks to the Author):

The authors have adequately addressed my concerns. The manuscript is much improved scientifically and stylistically.

Reviewer #2 (Remarks to the Author):

I appreciate the thorough approach the authors have taken in considering and addressing my comments. The immunometabolic mechanism and evidence is much stronger in the revised version. Congratulations on the paper.

Reviewer #3 (Remarks to the Author):

The revised manuscript looks much robust. The distinct regulation of IL-1b and IL-18 during CAPS and the predominant regulation of IL-1b (rather than IL-18) by DUBs certainly add strength to the reported findings.

Point-by-point response to reviewers' comments:

MS No.: NCOMMS-23-19097

MS Title: NLRP3 mutants pathogenic to Cryopyrin-Associated Periodic Syndrome form constitutively active inflammasome and override immune-metabolic inhibition of IL-1 β production

General: We are especially grateful to the reviewers for their comments. In the corrected version of our manuscript, editorial style changes are shown in word with the tracking tool.

Reviewer #1:

The authors have adequately addressed my concerns. The manuscript is much improved scientifically and stylistically.

Answer: *We appreciate the reviewer comments on the improvement of our study.*

Reviewer #2:

I appreciate the thorough approach the authors have taken in considering and addressing my comments. The immunometabolic mechanism and evidence is much stronger in the revised version. Congratulations on the paper.

Answer: We also appreciate the comments of the reviewer that have improve our study.

Reviewer #3:

The revised manuscript looks much robust. The distinct regulation of IL-1b and IL-18 during CAPS and the predominant regulation of IL-1b (rather than IL-18) by DUBs certainly add strength to the reported findings.

Answer: We appreciate reviewer comment.